**Influence of inherited structural domains and their particular strain distributions on the Roer Valley Graben evolution from inversion to extension**

Jef Deckers[1], Bernd Rombaut[1], Koen Van Noten[2] and Kris Vanneste[2]

1: VITO, Flemish Institute for Technological Research, Boeretang 200, BE-2400 Mol, Belgium
2: Seismology-Gravimetry, Royal Observatory of Belgium, Ringlaan 3, BE-1180 Brussels, Belgium

**ABSTRACT:**

The influence of strain distribution inheritance within fault systems on repeated fault reactivation is far less understood than the process of repeated fault reactivation itself. By evaluating cross-sections through a new 3D geological model, we demonstrate contrasts in strain distribution between different fault segments of the same fault system during its reverse reactivation and subsequent normal reactivation.

The study object is the Roer Valley Graben (RVG), a middle Mesozoic rift basin in Western Europe that is bounded by large border fault systems. These border fault systems were reversely reactivated under Late Cretaceous compression (inversion) and reactivated as normal faults under Cenozoic extension. A careful evaluation of the new geological model of the western RVG border fault system - the Feldbiss fault system (FFS) - reveals the presence of two structural domains in the FFS with distinctly different strain distributions during both Late Cretaceous compression and Cenozoic extension. A southern domain is characterized by narrow (< 3 km) localized faulting, while the northern is characterized by wide (>10 km) distributed faulting. The total normal and reverse throws in the two domains of the FFS were estimated to be similar during both tectonic phases. This shows that each domain accommodated a similar amount of compressional and extensional deformation, but persistently distributed it differently.

The faults in both structural domains of the FFS strike NW-SE, but the change in geometry between them takes place across the oblique WNW-ESE striking Grote Brogel fault. Also in other parts of the Roer Valley Graben, WNW-ESE striking faults are associated with major geometrical changes (left-stepping patterns) in its border fault system. At the contact between both structural domains, a major NNE-SSW striking latest Carboniferous strike-slip fault is present, referred to as the Gruitrode Lineament. Across another latest Carboniferous strike-slip fault zone (Donderslag Lineament) nearby, changes in the geometry of Mesozoic fault populations were also noted. These observations demonstrate that Late Cretaceous and Cenozoic inherited changes in fault geometries as well as strain distributions were likely caused by the presence of pre-existing lineaments in the basement.

## 1. Introduction

Rift basins are typically bounded by large fault systems. These border fault systems are generally segmented along strike. As they represent zones of pre-existing weaknesses, the large border fault systems are prone to reactivation under either extension or compression. The effects of pre-existing segmentation upon extensional or compressional strain distributions in reactivated rift border fault systems have thus far received little attention. One of the ideal areas to study these effects is at the border fault systems of the Roer Valley Graben (RVG). These systems developed in the middle Mesozoic, and were reversely reactivated under Late Cretaceous contraction and experienced normal reactivation again under Cenozoic extension (Demyttenaere, 1989; Geluk et al., 1994). The RVG border faults are dominantly NW-SE oriented, and locally intersected by WNW-ESE striking faults (Michon et al., 2003; Worum et al., 2005). Some of the largest WNW-ESE striking faults (such as the Grote Brogel, Lövenicher-Kast and Veldhoven faults) caused major left-stepping patterns in the overall NW-SE graben border geometry during compression as well as during extension. This is evidenced by gravimetric maps of the area (Fig. 1) and in more detail in maps of the middle Mesozoic (Jurassic), Upper Cretaceous and Cenozoic stratigraphic distributions and thicknesses in the area (c.f.

Duin et al., 2006; Deckers et al., 2019). This apparent influence of non-colinear (not in line) WNW-ESE striking faults on the development of the RVG border fault system through time has, however, never been studied in detail. This study aims at a better understanding of the role that inherited segmentation plays on later episodes of compressional and extensional graben border fault reactivation.

For site location, we selected the western border fault system of the RVG (Fig. 1), which in Flanders (northern Belgium) is characterized by long NW-SE faults (such as the Bocholt, Neeroeteren, Reppel and Rotem faults) and the major WNW-ESE oriented Grote Brogel fault (GBF). The Quaternary activity of the GBF and its influence on the local hydrology was recently studied at two investigation sites by means of shallow boreholes, Cone Penetration Tests, electrical resistivity tomography and geomorphic analysis by Deckers et al. (2018). To analyze the interaction of the GBF with the other faults in the western RVG and its influence on the large-scale graben geometry, we used recently published layer and fault models of the 3D Geological Model for Flanders (version 3; G3Dv3-model; Deckers et al., 2019) together with the digital elevation model. The G3Dv3-model of the area was created by the integration and interpretation of all available 2D seismic reflection and borehole data (borehole descriptions and wireline logs). It consists, among others, of stratigraphic layer and thickness maps for over hundred stratigraphic units ranging from the Quaternary at the surface to the Lower Paleozoic strata at depths of almost 10 km. These maps illustrate the Late Cretaceous and Cenozoic stratigraphic distributions with respect to the faults. Evaluating these maps allows reconstructing the geometrical changes of the study area through time.

## 2. Geological setting and stratigraphy

### 2.1 Paleo- and Mesozoic

The Brabant Massif, a relatively stable WNW-ESE trending continental block that consists of folded lower Paleozoic (Cambrian to Silurian) strata, is present throughout the subsurface of northern Belgium (Flanders). In the northeastern part of the Brabant Massif, the lower Paleozoic strata are covered by a thick (on average > 2000 m) wedge of upper Paleozoic (Devonian to Carboniferous) strata in an area referred to as the Campine Basin. The Carboniferous of the Campine Basin starts with a carbonate succession (Dinantian), transitioning to shales (Namurian) and ending in fluviatile successions of coal-rich claystone and sandstone alterations (Westphalian). The thickness distribution of Dinantian carbonates suggests syn-sedimentary normal fault activity with NW-SE to E-W strikes (Muchez & Langenaeker, 1993). Deformation of Westphalian strata in turn, points towards late Carboniferous block-faulting and tilting, partly along strike-slip faults (Bouckaert and Dusar, 1987). During this deformation phase (Saalian phase in Fig. 2), the roughly N-S trending Donderslag Lineament and NE-SW trending Gruitrode Lineament developed as transpressional structures in the southeastern part of the Campine Basin (Bouckaert & Dusar, 1987; Dusar & Langenaeker, 1992; Figs. 3 & 5). The Donderslag and Gruitrode lineaments are expressed as anticlines in the Westphalian strata with maximum amplitudes of about 500 m (Rombaut et al., 2020). The deformed Westphalian strata were unconformably overlain by latest Permian and Triassic continental to shallow marine successions.

From the latest Triassic onwards (Early Cimmerian phase in Fig. 2; Geluk et al., 1994), fault activity was noted along a large number of predominantly NW-SE and WNW-ESE striking faults across the area (Worum et al., 2005). This activity resulted in differentiation of the Paleozoic Campine Basin into several major tectonic blocks. The RVG was the strongest subsiding block, flanked by the Campine Block (CB) in the west and the Peel Block in the east. Probably during the latest Jurassic (Late Cimmerian phase in Fig. 2), the entire region was uplifted and most of the syn-rift strata were eroded outside and also locally within the RVG (Fig. 3). For the purpose of this study, the Jurassic and older strata will be referred to as the pre-Cretaceous strata.

During subsequent Late Cretaceous (Campanian to middle Maastrichtian) compression, referred to
as the Sub-Hercynian phase, the Campine and Peel Blocks experienced subsidence with the
deposition of generally between 200 and 300 m of carbonates of the Chalk Group, while the RVG in
between them was squeezed upwards or inverted (Geluk et al., 1994; Figs. 2 & 3). Inversion of the
RVG took place by reverse movements along its (pre-existing) border faults (Demyttenaere, 1989).
Apatite fission track analyses revealed that the amount of late Cretaceous uplift of the RVG is
remarkably similar to the amount of subsidence of its flanks (Luijendijk et al., 2011). Inversion in the
area probably took place under a N-S to NNW-SSE direction of maximum horizontal compression (de
Jager, 2003) as the result of convergence between Africa and Europe (Kley and Voigt, 2008). A sharp
decrease in the convergence rates between Africa and Europe during the latest Maastrichtian
(Rosenbaum et al., 2002) ended the Sub-Hercynian phase in the region. This is evidenced by the
widespread deposition of the youngest (uppermost Maastrichtian and Danian) sequence of the Chalk
Group, which is also present on top of formerly inverted basins (Deckers & Van der Voet, 2018). Our
informal definition of the Chalk Group, however, only contains those parts of the Chalk Group that
were deposited during inversion of the RVG, which are missing inside the RVG. The uppermost
Maastrichtian and Danian sequences are therefore not included in the information Chalk Group
definition in this study.
**2.2 Cenozoic**
From the early Cenozoic onwards, the study area was situated in the southern part of the North Sea
Basin and covered by several hundreds of meters of siliciclastics (Fig. 2). Some tectonic phases did
occur between the start of the Paleogene and the end of the early Oligocene (Fig. 2), but without
major fault activity (c.f. Deckers & Van der Voet; 2018). For the purpose of this study, the latest
Maastrichtian to early Oligocene strata are referred to as the pre-rift strata.
Major fault activity resumed in the late Oligocene, when the Roer Valley Rift System developed as a
northwest-trending branch of the Rhine-Graben-System (Ziegler, 1988), throughout the south-
eastern part of the Netherlands, eastern Belgium and adjacent parts of Germany (Fig. 1). This system
currently extends over a distance of roughly 200 km and has a width of up to 75 km. The faults with
the strongest displacements divide the central Roer Valley Rift System into the Campine Block in the
west, the pre-existing Roer Valley Graben in the center and Peel Block in the east. The Roer Valley
Rift System is currently still active as indicated by the earthquake activity in the region (Fig. 1). Syn-
rift sedimentation started in the late Oligocene with the deposition of the Voort Formation (base
syn-rift strata in this study; Fig. 2). After the Oligocene, sedimentation gradually coarsened from
shallow to marginal marine glauconitic sands (Bolderberg and Diest formations; clinoforms in Fig. 6)
until the end of the Miocene, to coarser marginal marine to fluvial sands in the Pliocene (Mol and
Kieseloolite formations) and gravel-bearing fluvial sands in the Quaternary (Meuse Group; Fig. 2).
Due to the relatively strong resistance to erosion of the gravel-bearing sands of the Meuse Group,
the easternmost part of the Campine Block is currently a relatively high area (often referred to as
the Campine Plateau; Fig. 4) delimited to the west by the deposition limit of these coarse sediments
and in the east by the major border faults of the RVG (Beerten et al., 2013; Verbeeck et al., 2017),
which separate the Campine Plateau from the Reppel, Kaulille and Bocholt Plains (Paulissen, 1997,
Fig. 4). As a result of continuous rifting since the late Oligocene, the abovementioned stratigraphic
units are relatively thick in the RVG (over 1000 m) compared to the flanking CB and Peel Blocks
(generally below 500 m; Demyttenaere, 1989; Geluk, 1990; Fig. 3).
During Miocene to recent rifting, fault distribution in the Roer Valley Rift System was characterized
by two main trends: the dominant NW-SE (N145-160) trend corresponding to the general orientation
of the graben, and the secondary WNW-ESE (N110-120) oblique orientation (Michon et al., 2003).
These directions were both favorable for fault reactivation under the NE-SW Miocene to recent
extensional direction (Michon et al., 2003; Michon & Van Balen, 2005). Along its eastern border, the
RVG is separated from the Peel Block by the Peel Boundary fault zone, a NW-SE oriented, 100 km
long narrow deformation zone composed of the Peel boundary fault and several secondary faults
(Michon & Van Balen, 2005; Fig. 1) with a total vertical throw of 400-800 m for the base of the
Miocene (Geluk et al., 1994). Along its western border, the RVG is separated from the CB by a broad
fault bundle, the Feldbiss fault system (FFS), which consists of a number of faults showing a left-
stepping pattern (Fig. 1). As a result of this left-stepping pattern, the RVG changes from a near full
graben in the center  to an asymmetric graben in the north (Michon & Van Balen, 2005; Fig. 1). The
FFS is 80 km long and is mainly composed of the Feldbiss fault, the Geleen (NL) or Neeroeteren (BE)
fault and the Heerlerheide (NL) or Rotem (BE) fault (Michon & Van Balen, 2005; Fig. 4) and shows
vertical throws of the base of the Miocene of roughly 400 m (Demyttenaere & Laga, 1988). The
stratigraphic thicknesses indicate that the Peel Boundary fault system was generally more active than
the FFS since the beginning of the Miocene (Michon & Van Balen, 2005; Fig. 3). Consequently, the
main Miocene to recent depocenters developed in the hangingwall of the Peel Boundary fault
system.
The study area is centered on the GBF, which is situated in the central portion of the FFS (Fig. 1). The
GBF branches off from the major Neeroeteren fault in a WNW-ESE orientation. It has a pronounced
geomorphic scarp (up to 4m) which gradually fades away towards the west (Fig. 4). This gradual
disappearance coincides with the decrease in fault throw of the Pleistocene Meuse terraces (Deckers
et al., 2018).
**3. Dataset and methodology**
**3.1 General dataset**
In the past two decades, a large number of (hydro)geological models have been created for the study
area (c.f. Langenaeker, 2000; Sels et al., 2001; Beerten et al., 2005; Meyus et al., 2005; Matthijs et
al., 2013; Deckers et al., 2019) or parts of it (Deckers et al., 2014; Vernes et al., 2018). For the purpose
of this study, we rely on the most recently published 3D subsurface model of Flanders, called the
G3Dv3-model (Deckers et al., 2019). This model consists of 3D models of over hundred
lithostratigraphic units from the Lower Paleozoic (at depths of up to 10 km) up to the Quaternary at
the surface. The G3Dv3-model also contains 3D surfaces of over two hundred faults in the eastern
part of Flanders. For the eastern border region between Flanders and the Netherlands, the
3D(hydro)geological models of the Cenozoic stemming from two cross-boundary projects, namely
the H3O-Roer Valley Graben and H3O-Campine area (Deckers et al., 2014; Vernes et al., 2018), were
integrated and stratigraphically further detailed/updated in the G3Dv3-model. Consequently, the
G3Dv3-model combines the most recent geological knowledge in Flanders.
The main data sources to create the G3Dv3-model were the following:
o Boreholes: Several tens of thousands of borehole descriptions from Flanders are present in the
databases of DOV ("Database subsoil Flanders"; https://www.dov.vlaanderen.be/) and of the
Geological Survey of Belgium. Besides the descriptions, these databases often contain one or
more interpretations of the lithostratigraphic successions (groups, formations, members) in each
borehole. Thousands of these interpretations were selected from these databases as a starting
point to create the geological models. After selection, the existing lithostratigraphic
interpretations of the boreholes were critically examined and accepted, rejected or
reinterpreted for the different stratigraphic layers. An overview on the used boreholes to map
the Chalk Group, base syn-rift strata and base Quaternary and Pliocene strata is shown in figure
200   5.
o Seismic data: The eastern part of Flanders is covered by over 400 lines from numerous seismic
campaigns that were performed between 1953 and 2015. This dataset consists of more widely
spaced seismic lines from a regional seismic survey performed between 1953-1956 (Campine
Basin), complemented by dense networks of more closely spaced lines from local surveys mainly
conducted between the 1980's and 2015. In general, the quality of the seismic data improves
with time. Besides the age, also the targeted depth-range of the seismic survey strongly
influences the vertical resolution of the seismic data. Some seismic surveys target deep (> 2 km)
Lower Carboniferous strata, while others target shallow (< 1 km) Cenozoic strata. Consequently,
the quality of the resulting image is better for the deep and shallow range, respectively. The
entire selection of seismic lines was interpreted for horizon and fault mapping. An overview on
the interpreted seismic lines to map the Chalk Group and base syn-rift strata is shown in figure
5.
o Topographic data: The topography forms the top of the G3Dv3-model. This topography was
constructed mainly from the Digital Terrain Model of Flanders (DTMV-II) from Agentschap
Informatie Vlaanderen (2018). As a result of their recent activity, several of the large RVG
boundary faults are expressed in the topography by relief gradients or scarps (Camelbeeck &
Meghraoui, 1996; Paulissen, 1997; Fig. 4). The relief gradient provides a good indication of the
location and orientation of the fault traces of these boundary faults at the surface, especially
when used in combination with the seismic data.
**3.2 Dataset in the study area**
The dataset used to analyse the study area is limited to a horizontal (area around the GBF) and
vertical (depth interval corresponding to Upper Cretaceous and Cenozoic) subset of the G3Dv3-
model. Because of this restriction the following data-selection was made:
Stratigraphic layers:
o The top, base and thickness of the Upper Cretaceous syn-inversion strata of the Chalk Group
were selected, since they illustrate the inversion-related (Late Cretaceous) deformation. The
top and base of the Chalk Group were mainly based on seismic interpretations, locally
supported by borehole interpretations (for location, see Fig. 5). Boreholes and seismic data
show that the Chalk Group is absent within the RVG, and up to 300 m thick in the CB (Fig. 3).
o The bases of the syn-rift (Voort Formation) and Pliocene to Quaternary strata (equivalent
Mol/Kieseloolite Formations) were selected to illustrate the Cenozoic extension-related
deformation. The model of the base syn-rift strata was mainly based on seismic
interpretations either from this horizon itself or from a nearby horizon, and supported by
borehole interpretations (for location, see Fig. 5). The base of the Pliocene to Quaternary
strata is generally too shallow to be consistently seismically interpreted, and was therefore
based on borehole interpretations (for location, see Fig. 5). Due to the shallow location, the
number of available boreholes for the Pliocene to Quaternary strata was high compared to
those available to map the underlying layers (compare in Fig. 5).
Faults:
From the sets of faults in the G3Dv3-models in the study area, we only selected those that show an
offset in the Chalk Group and in the base of the syn-rift strata (see Figs. 7 & 9). Due to the relatively
large spacing between the boreholes, predominantly seismic data were used for fault mapping. Since
all of the used seismic data are two-dimensional, only fault lines are imaged and their lateral
connection into one fault plane remains interpretative. The long faults discussed in this study should
therefore not be considered as single fault planes, but rather as fault systems, each of which
represents one tectonic feature made up by different fault lines that can represent either linked or
isolated fault segments (following Rypens et al., 2004). The interpreted lateral connection of the 2D
fault lines into 3D fault systems was predominantly based on the comparison of the variation of
vertical displacements between adjacent seismic profiles, locally supported by topographic
indications and borehole data. The most reliable fault models are therefore created from areas with
low structural complexity, high seismic coverage, large numbers of boreholes and strong topographic
expression of the faults.
2D seismic coverage is generally high for the area south of the village of Bree because of the dense
networks of different seismic surveys across the RVG border fault system (Fig. 5). The seismic
interpretations and lateral connections of the border faults (such as the Bree, Dilsen, Neeroeteren
and Rotem faults) are therefore most reliable in this area. In addition, the major Neeroeteren fault
is clearly expressed in the topography as a large (+/- 30m) relief gradient, often referred to as the
Bree Fault Scarp (Camelbeeck & Meghraoui, 1996; Fig. 4).
North of the village of Bree, on the other hand, 2D seismic coverage is very poor with only five long,
low to average quality seismic lines (either old or only imaging the Cenozoic; Fig. 5). Consequently,
interpreting faults and their lateral connections in the RVG border zone on seismic data alone would
have a high degree of uncertainty. The southern sections of the Bocholt, GBF and Reppel faults are,
however, clearly expressed by topographic gradients (Fig. 4), which provide support for the seismic
fault line connections. At locations where the topographic expression fades, however, the
uncertainty increases again:
o   In its eastern portion, the GBF is clearly expressed by a topographic gradient of over 10 m
with a clear fault scarp up to 4 m high near Bree (Deckers et al., 2018; Fig. 4). As its throw
decreases in western direction, however, its topographic expression fades, which causes a
major uncertainty (of several km's) on the exact location and extent of the northwestern tip
of the GBF.
o   In the western portion of the GBF, Demyttenaere & Laga (1988) interpreted an important
bend towards the NW-SE Overpelt fault (Fig. 5). The topographic expression is, however, too
faint to corroborate this bend in the GBF (Fig. 4). A recently reprocessed seismic line nearby
also shows only a minor throw near the location of the supposed bend (question mark on
Fig. 5). So although this bend of the GBF towards the Overpelt fault is indicated as a major
fault in geological models, its importance remains largely uncertain and is therefore
indicated on figures 5, 7 and 9 with question marks. Contrary to the bend of the GBF, the
presence of the Overpelt fault is supported by several seismic lines (Figs. 5 & 6). The Overpelt
fault runs more or less parallel to the Reppel and Bocholt-Hamont faults further east.
o   Due to the lack of clear topographic expression of faults and due to the diffuse seismic
coverage, a large uncertainty remains on fault interpretations in the area between the
Overpelt fault and the Rauw fault 14 km further west. West of the Rauw fault, the seismic
coverage increases again and the uncertainty on fault interpretations and their lateral
connection decreases.
Palaeozoic Lineaments:
For the G3Dv3-model, the trace of the latest Palaeozoic Donderslag Lineament was interpreted on
the 2D seismic data. In a later modelling of the uppermost Carboniferous strata, also the latest
Palaeozoic Gruitrode Lineament was interpreted and modelled by means of seismic and borehole
data (Rombaut et al., 2020). The expression of the Gruitrode Lineament as an anticline on seismic
data is shown in figure 3. The traces of the Donderslag and Gruitrode Lineaments are shown in figure
5.
**3.3 Methodology**
To illustrate the Cenozoic syn-rift geometry in the study area, we show an ArcGIS map view of the
G3Dv3-model of the depth of the base of the syn-rift strata and the affecting faults in Figure 5. An

overview of the total vertical throw at the base of the syn-rift strata along some of the major faults of the FFS is shown in figure 8.

To illustrate the Late Cretaceous syn-compressional geometry in the study area, we show an ArcGIS map view of the G3Dv3-model of the thickness of the Chalk Group and the major faults that are known to have influenced it in figure 9.

Besides the map views, also four cross-sections (Figs. 10A to -E) of the G3Dv3-model were constructed (by means of iMOD software) to illustrate the Late Cretaceous to recent sediment thicknesses and geometries in the study area. Three cross-sections are SW-NE oriented, perpendicular to the graben trend (10A, -B and -C) and two others SE-NW oriented, sub-parallel to the graben trend (10D and -E). On these cross-sections, the top and base of the Chalk Group and pre-rift strata are indicated. The syn-rift strata are divided in two parts, namely the late Oligocene and Miocene below and the Pliocene to Quaternary on top.

## **4. Results**

### **4.1 Structural style of Cenozoic rifting**

The model of the base of the syn-rift strata (base upper Oligocene) illustrates the geometry of Cenozoic rifting. In the RVG, the base of the syn-rift strata is currently situated in the subsurface at depths that generally exceed -900 m TAW (Tweede Algemene Waterpassing; Fig. 7). In the CB, the base of the syn-rift strata is located at more shallow depths, ranging from at +50 m TAW in the southwest up to -400 m TAW in the northeast (Fig. 10). Towards the easternmost parts of the CB, this trend becomes progressively more disturbed by the presence of faults and tilted blocks in the footwall domain of the FFS (Fig. 7). For individual faults in the CB, the vertical throws at the level of the base of the syn-rift strata do not exceed 80 m.

At the FFS, the base of the syn-rift strata drops by 500 m from the CB into the RVG (from -400 m TAW to -900 m TAW; Fig. 7). This jump takes mainly place by - often large - vertical throws along a dense, complex network of normal faults of the FFS, and in between those also by an eastward dip of the syn-rift strata (Figs. 3, 6 & 10). Several faults in the FFS have vertical throws of the base of the syn-rift strata of over 150 m (Bocholt, GBF, Hamont, Overpelt, Reppel, Rotem), with a maximum of almost 600 m along the Neeroeteren fault (Fig. 8).

In the RVG itself, vertical fault throws are larger than in the CB, but smaller than in the FFS as they generally do not exceed 150 m (Figs. 3 & 7). Most of the intra-graben faults are dipping in the direction of the nearest graben border fault system (i.e. are antithetic; Figs. 10A and -B). The simultaneous activity of the synthetic graben border faults and the antithetic intra-graben faults resulted in a series of long subgrabens in the western flank of the RVG (Fig. 7; Deckers, 2016).

The geometry of the FFS shows strong lateral changes across the study area. Within the FFS, two structural domains and their particular geometry were identified, north and south of the village of Bree (for location, see Fig. 4):

- o The southern domain consists of the NW-SE Neeroeteren and Rotem faults (Figs. 7 & 8). The width of this domain is limited to the Neeroeteren fault in the north, and from the branching point with the Rotem fault onwards increasing in southern direction up to a maximum of 2 km near the Belgian/Dutch border. Most of the vertical throw of the FFS is taken by the NW-SE striking Neeroeteren fault, with vertical throws of the base of the syn-rift strata of almost 600 m (Figs. 7, 8 & 10A and -B). The Rotem fault shows a maximum vertical throw of the base of the syn-rift strata of 150 m (Figs. 7, 8 & 10A). The large vertical throw along the Neeroeteren fault is also expressed by a strong relief gradient denoted as the Bree Fault Scarp on top of this fault (topographic offset between 15-20 m; Fig. 4). This relief gradient coincides with the boundary between the elevated (> 50 m TAW) Campine Plateau on top of the CB and the low-lying (< 40 m TAW) Bocholt Plain on top of the RVG (c.f. Paulissen, 1997;

Fig. 4). The relief gradient of the Bree Fault Scarp is evident between Bree and the hamlet of
Waterloos, but abruptly disappears south of Waterloos due to the WSW-ENE incision by the
Quaternary Meuse river from the late Pleistocene onwards (Fig. 4).
o  The northern domain starts where the Neeroeteren fault bifurcates into the GBF towards
the west and the Bocholt fault towards the north (Figs. 7 & 8). These two faults define the
boundaries of the northern domain of the FFS. Since the GBF and Bocholt faults have a
WNW-ESE and NW-SE strike respectively, the northern domain progressively widens in
northern direction, reaching a width of up to 13 km in the central part (Fig. 10C). As it
bifurcates, the large vertical throw along the Neeroeteren fault (about 530 m) is roughly
equally divided over the GBF and Bocholt faults (about 280 m and 220 m respectively; Fig.
8). As a result, while the southern domain delimits a high footwall area in the west from a
low hangingwall area in the east, the northern domain shows a more gradual downfaulting
in eastern direction, with relatively small throws across some of its major faults (compare
Figs. 3 & 6). The smaller vertical throws along faults in the northern domain is also expressed
by absent or only very small relief gradients for most of its faults (excluding the GBF; Fig. 4).
As one of the most important faults, the Bocholt fault for example shows a vertical
topographic offset of maximum 4 m near the town of Bree where the fault is only expressed
as a low angle linear slope without a clear scarp. Consequently, while the Bocholt Plain and
Campine Plateau are clearly delimited by the Bree Fault Scarp in the southern domain, their
transition is much more stepwise along smaller fault scarps in the northern domain (Fig. 4).
This stepwise topography has led to the subdivision of the Lommel, Reppel and Kaulille Plains
in the northern domain of the FFS and its hangingwall (Paulissen, 1997; Fig. 4). The GBF forms
the boundary between the elevated Campine Plateau and the lower Reppel and Kaulille
Plains and consequently has a large topographic relief in respect to the NW-SE striking faults
in the northern domain. This topographic relief is largest in the east (15-20 m) where it seems
to be in continuation with the Bree Fault Scarp associated with the Neeroeteren fault
(Deckers et al., 2018). As the total vertical throw along the GBF decreases in western
direction, its relief gradient also decreases (Fig. 4). This decrease is, however, not gradual.
Deckers et al. (2018) noticed an abrupt decrease in the topographic throw at about 2 km
west of the eastern tip of the GBF. These authors related this decrease to the Reppel fault
branching off from the GBF, taking over part of the total displacement (Fig. 4). Also at depth,
the large vertical throw of the base of the syn-rift strata along the GBF of 270 m abruptly
decreases towards 170 m across the contact point with the Reppel fault (Fig. 8). This
decrease of 100 m in vertical throw along the GBF is completely accommodated by the
vertical throw of 110 m along the Reppel fault (Fig. 8). West of the bifurcation with the
Reppel fault, vertical throw along the GBF decreases further towards 100 m (Fig. 7). As
mentioned above, at the western tip of the GBF, several authors have previously suggested
a bend towards the NW-SE Overpelt fault (Demyttenaere & Laga, 1988; Broothaers et al.,
2012; Deckers et al., 2015). What is clear from the seismic data is the presence of the NW-
SE striking Overpelt fault further north with vertical throws in the order of 100 m (Fig. 6).
Northwest of the village of Peer, however, the topographic expression of the GBF becomes
faint and there is no further data (borehole nor seismic) to provide indications on its
westward continuation (Fig. 4). The decrease of topographic relief in north-western direction
coincides with the decrease of vertical throw of the syn-rift strata along the large faults in
the northern domain (Fig. 7). Subsidence from the CB towards the RVG is still partly
accommodated by small faults but increasingly by a strong northeastward dip of the base of
the syn-rift strata (Figs. 7 & 10C).
The cross-section of figure 10E (or RVG) illustrates that no major changes in thickness of the syn-rift
strata take place from the hangingwall of the southern domain of the FFS towards the hangingwall
of the northern domains of the FFS. The map view of figure 7 shows that this is also the case in the
footwall of the FFS (or CB). This shows that the total throw of the FFS does not strongly change across
the boundary between both domains.
**4.2 Structural style of Late Cretaceous inversion**
Under Late Cretaceous compression, the CB and Peel Blocks were downthrown, while the RVG in
between them was pushed upwards or inverted. Consequently, the late Cretaceous Chalk Group is
absent in the RVG and currently up to 300 m thick within the CB (Figs. 3, 6 & 9). Uplift of the RVG
was accommodated by reverse movements along its border faults, i.e. the FFS. As the Chalk Group
was deposited in the CB during inversion of the RVG, the thickness changes of the Chalk Group across
the FFS provide an indication on the (minimum) amount of total reverse throws along the FFS. Since
the Chalk Group is about 250 to 300 m thick in the footwall of the FFS (or CB) and absent in the
hangingwall of the FFS (or RVG), the total amount of uplift along the FFS can be estimated at over
300 m (taking into account later compaction of the chalks). This amount of uplift is consistent with
the range (250 to 500 m) obtained from apatite fission track measurements by Luijendijk et al. (2011)
in the nearby borehole Nederweert. Thickness changes of the Chalk Group across individual faults of
the FFS can also be used for quantification of the syn-inversion reverse movements along these
faults. In the eastern section of the FFS the Chalk Group is, however, absent. Therefore, only reverse
movements of the westernmost faults within the FFS can be reconstructed. The thickness maps of
the Chalk Group indicate different structural patterns of uplift across the western faults of the FFS.
Similar to the Cenozoic (section 4.1), these differences can be separated geographically into a
southern and northern domain:
o   In the part of the CB south of the town of Bree (for location, see Fig. 4), the thickness of the
Chalk Group generally increases in the direction of the RVG to reach a maximum of almost
300 m in the footwall of the FFS (Fig. 9). From this footwall, the Chalk Group strongly thins
across reverse faults. Three major reverse faults were observed, namely the Bree, Rotem
and Dilsen faults. The Chalk Group has a thickness of 250 m in the footwall of these faults,
becoming less than 150 m thick in the hangingwall of the Dilsen fault (Fig. 10A), and very thin
or even absent in the hangingwall of the Bree and Rotem faults (Figs. 3, 10A and -B). This
shows that vertical reverse throws along faults reached 100 to 250 m or more. Since the
Dilsen fault is present in the footwall and converges (in the Palaeozoic basement) towards
the more important Rotem fault, the Dilsen fault may represent a footwall shortcut fault of
the Rotem fault. In a similar manner, the Bree fault may also represent a footwall shortcut
fault of the Neeroeteren fault, although the convergences of the first towards the latter is
not obvious on seismic data (Fig. 3). If they indeed represent footwall shortcut faults, the
Bree and Dilsen faults would have originated during Late Cretaceous compression to
accommodate inversion on the pre-existing Neeroeteren and Rotem faults. This hypothesis
is supported by the fact that the base of the Lower to Middle Mesozoic strata shows a very
similar amount of reverse vertical throw as the base of the Chalk Group along the Bree fault
(Fig. 3). Nevertheless, earlier (Cimmerian) activity along the Bree and Dilsen fault cannot be
excluded. Contrary to most other faults in the FFS, the Bree and Dilsen faults were not
reactivated during Cenozoic extension and therefore now still have net reverse throws (Figs.
3, 10A and -B).
o   North of the town of Bree (for location, see Fig. 4), from the Rauw fault onwards, the Chalk
Group thins in eastern direction until it becomes absent near the Overpelt fault (Figs. 6 &
10C). East of the Overpelt fault, the Chalk Group is absent and reverse fault throws are
unknown. The zone along which the Chalk Group thins is therefore at least 10 km wide.
Contrary to the southern domain, thinning of the Chalk Group is not very abrupt across major
reverse faults or thrust faults in the northern domain. Instead, it takes place by small reverse

displacements along faults and predominantly by upwards flexuration (see flexures at the
top and bottom of the Chalk Group in Figs. 6 & 10C) towards the northeast.
The transition between the southern and northern domains is located at or along the lateral extent
of the GBF (Fig. 9). The Chalk Group is about 200 m thick in the footwall of the GBF (the CB), but
absent in its hangingwall (the northern domain of the FFS), which indicates that this fault had a
reverse throw of at least 200 m (Fig. 10D). This throw decreases in western direction along the GBF
(Fig. 9). Northwest of the western tip of the GBF, northwards thinning of the Chalk Group takes
mainly place by upwards flexuration of the basement (Figs. 6 & 10C).
**5. Discussion**
**5.1 Graben border activity and segmentation**
Based on stratigraphic maps extracted from the new 3D geological model of Flanders (G3Dv3-model;
Deckers et al., 2019), it is shown that, in agreement with former studies (Rossa, 1986; Demyttenaere
& Laga, 1988; Demyttenaere, 1989; Langenaeker, 2000), the FFS was highly active during both Late
Cretaceous contraction and Cenozoic extension. Like many other faults in the region, the FFS
probably developed during the middle Mesozoic Early Cimmerian phase (Fig. 2). The FFS thereby
enabled the structural differentiation of the RVG in its hangingwall from the relatively high CB in its
footwall. Due to erosion during the late Cimmerian phase (Geluk et al., 1994; Fig. 2), Jurassic syn-rift
strata are not preserved in the CB and only locally in the RVG (Fig. 3), which makes it difficult to
reconstruct the early Cimmerian fault kinematics. However, fault-related deformation and fault-
bounded preservation of Triassic and Lower Jurassic strata indicate that most faults in the FFS were
active during the Early Cimmerian phase.
Under Late Cretaceous compression, the RVG was inverted by reverse reactivation of the middle
Mesozoic FFS. As the result, the RVG is lacking an Upper Cretaceous cover, while the neighboring CB
was covered by 200-300 m of Upper Cretaceous carbonates of the Chalk Group (Figs. 3, 6, 9 & 10).
The simultaneity of inversion of the RVG and subsidence of the CB in its flank is evidenced by the
progressive increase of clastic sediment input in the Chalk Group in the direction of the RVG (Bless
et al., 1986). The thickness maps of the Chalk Group in this study provide no indication for important
Late Cretaceous fault activity in the CB (Fig. 9). Late Cretaceous compressional strain distribution was
therefore fundamentally controlled by and focused on the pre-existing FFS. The focus of strain upon
the FFS might be the result of the large size it had reached during middle Mesozoic rifting, since
large-sized faults have the potential to accrue displacement at the expense of smaller-sized
surrounding faults (Reilly, 2017). Based on the stratigraphic thickness analyses of the Chalk Group in
the CB in this study and apatite fission track analyses in boreholes in the RVG by Luijendijk et al.
(2011), total Late Cretaceous reverse throws along the FFS are estimated to be in the order of 300
m. The interpreted seismic lines (Figs. 3 & 6), map views (Fig. 9) and cross-sections (Fig. 10) indicate
that this reverse throw of the FFS was accommodated by reverse vertical throws along individual
faults (up to 250 m) as well as by upwards flexuring of the pre-Chalk Group basement.
Under Cenozoic (late Oligocene to recent) extension, the middle Mesozoic faults were again
reactivated (Demyttenaere, 1989). Contrary to the Late Cretaceous contraction, Cenozoic normal
reactivation was not limited to the FFS, since also numerous surrounding faults in the CB and RVG
were reactivated in a normal movement (Figs. 3, 7 & 10). The map of the base of the Cenozoic syn-
rift strata (Fig. 7) shows that the majority of the extensional strain was again focused onto the FFS.
The interpreted seismic lines (Figs. 3 & 6) and cross-sections (Fig. 10) show that the FFS is
characterized by a relatively high concentration of East-dipping faults with vertical offsets that
exceed 150 m. The FFS thereby forms the transition from the relatively high CB (base syn-rift strata
> -400 m TAW) towards the low RVG (base syn-rift strata < - 900 m TAW). As a result of their normal
reactivation, the faults within the FFS experienced a reversal from Late Cretaceous reverse faulting
towards late Cenozoic normal movements. Some faults thereby reached a net normal offset at the
level of the base of the Late Cretaceous (Rotem fault; Fig. 10A), while others retain a net reverse
offset at the same stratigraphic level (GBF; Fig. 10D).
However, not all of the active Late Cretaceous faults in the FFS were reactivated during the Cenozoic.
The Bree and Dilsen faults (Figs. 3, 10A and -B), for example, are important Late Cretaceous faults
(reverse throws > 100 m) in the footwalls of the Neeroeteren and Rotem faults that were not
reactivated during the Cenozoic. Contrary to the other faults, the Bree and Dilsen faults might
represent footwall shortcuts to accommodate Late Cretaceous inversion of the larger Neeroeteren
and Rotem faults. The presence of footwall shortcut thrusts is characteristic for the early stages of
inversion of extensional fault systems (McClay & Buchanan, 1992). The lack of Cenozoic reactivation
of the Bree and Dilsen faults, contrary to the surrounding faults of the FFS, may relate to the Late
Cretaceous origin as thrust faults of the former two compared to the middle Mesozoic origin as
normal faults of the last ones. The middle Mesozoic major normal faults (Neeroeteren and Rotem),
rather than their footwall shortcut thrust faults (Bree and Dilsen), thus appear to have been more
preferable sites for the accommodation of the Cenozoic extension strain.
Maps of the Upper Cretaceous and Cenozoic stratigraphic distributions and thicknesses (Figs. 7 & 9)
indicate that the geometry of the FFS shows major lateral changes into distinct structural domains.
We identified two of those structural domains in the Belgian sector of the FFS, which existed during
both Late Cretaceous contraction and Cenozoic extension:
- In the southern domain, the FFS is characterized by a narrow (< 2 km) border fault zone
that is dominated by localized faulting. During the Late Cretaceous compression, this domain
was dominated by several faults with reverse throws of over 200 m, and during Cenozoic
extension by the large Neeroeteren fault with a normal throw of almost 600 m.
- In the northern domain, the FFS is characterized by a wide (average >10 km) border fault
zone that is dominated by distributed faulting. During the Late Cretaceous compression this
domain was characterized by upwards flexuring of the pre-Chalk Group basement and faults
with generally small reverse throws, and during Cenozoic extension by downwards tilting of
the pre-rift strata and faults with total throws of less than 250 m.
The northern and southern domains thus show persistent distributed and localized strain,
respectively, during phases of both contraction (Late Cretaceous) and extension (Cenozoic). Faults
with the largest Late Cretaceous reverse throw therefore also had the largest Cenozoic normal
throw. Mora et al. (2009) showed that for the Eastern Cordillera of Colombia the same is true for
faults that have been reactivated in the opposite way (i.e. faults with the largest normal throws also
showed the largest reverse throw during reactivation). The width of deformation, the degree of
shortening, the spatial development of structures, and the focus of ongoing tectonic activity seems
to be fundamentally influenced by the inherited structures (Mora et al., 2006). The similarity of the
geometry of the different domains of the FFS during both Late Cretaceous inversion and Cenozoic
extension shows that inherited structures also controlled the evolution of the border zone of the
RVG. This emphasizes the importance of pre-existing structural domains on tectonic deformation
during both inversion and extension, besides more obvious factors such as the fault strikes with
respect to the stress-field orientations. Indeed, under Late Cretaceous contraction and Cenozoic
extension, strain distribution remained similar in both structural domains of the FFS, while maximum
horizontal stress was estimated to be N-S to NNW-SSE for the first phase (de Jager, 2003) and NNE-
SSW to NW-SE for the latter phase (Michon et al., 2003; Michon & Van Balen, 2005).
**5.2 Possible cause for segmentation**

The abovementioned geometrical changes in the FFS had no influence on its total Cenozoic vertical
throw, which remains in the order of 600 m across both domains (Figs. 7 & 10E). They also seem to
have had little effect upon the thicknesses of the Upper Cretaceous Chalk Group, which remains in
the order of 250-300 m across the footwall of both domains of the FFS (Fig. 9). This shows that a
similar amount of strain (extensional and compressional) was distributed differently between the
southern and northern domains of the FFS, namely localized in the first and distributed in the latter.
Localized and distributed regimes of faulting are known to have occurred within one fault system
(Soliva and Schultz, 2008; Nixon et al., 2014), similar to the FFS in this study. Differences between
these regimes are often attributed to the maturity of the fault systems (Nixon et al., 2014): highly
mature systems are linked and have localized strain, whereas younger (less mature) faults are less
linked and more diffuse. In our study area, however, the difference in strain localization cannot be
related to differences in maturity, since the FFS is a long-lived system, already active since the middle
Mesozoic. Alternatively, such as is the case in the East African Rift System, the difference in fault
localization could be attributed to the presence of magmatic intrusions (Ebinger and Casey, 2001;
Kendall et al., 2005; Wright et al., 2006) or oblique pre-existing shear zones (Katumwehe et al., 2015;
Dawson et al., 2018). The RVG is, however, considered amagmatic during the Late Cretaceous and
Cenozoic, and pre-existing shear zones were up to very recently not known at the junction between
the two domains. The nearest known shear zone is the Gruitrode Lineament, a latest Carboniferous
dextral transpressional flexure zone (Bouckaert & Dusar, 1987) that runs NE-SW in the CB west of
the Bree Uplift (Langenaeker, 2000; Figs. 3 & 5). In a recent 3D modelling campaign of the latest
Carboniferous strata, the strike of the Gruitrode Lineament was revised into a NNE-SSW orientation
(Rombaut et al., 2020; for location see Fig. 5). As a result, the new trace of the Gruitrode Lineament
cuts the FFS at the junction between the southern and northern domain of this study. The Gruitrode
Lineament does not seem to continue east of the FFS or in the RVG (Rombaut et al., 2020; Fig. 5).
This shows that some of the faults in the FFS (at least the Neeroeteren and Bocholt faults) influenced
activity along the Gruitrode Lineament. The Neeroeteren and Bocholt faults are part of the
population of NW-SE striking faults, some of which were already active early in the Carboniferous
(Muchez & Langenaeker, 1993), and could therefore indeed have played an important role during
the latest Carboniferous formation of the Gruitrode Lineament. During middle Mesozoic rifting, the
same NW-SE striking faults were reactivated again and became, in the case of the Neeroeteren and
Bocholt faults, part of the larger FFS in the border of the RVG. The Gruitrode Lineament on the other
hand was not reactivated after the Paleozoic, as evidenced by the lack of deformation in its
overburden (Bouckaert & Dusar, 1987). Because of its position at the boundary between the
different domains, the Gruitrode Lineament did, however, play an important role in segmentation of
the FFS into the abovementioned two domains. Also further west in the CB, the middle Mesozoic
fault pattern is known to have been influenced by another - predominantly NNE-SSW-striking
(Deckers et al., 2019) - latest Carboniferous transpressional flexure zone, called the Donderslag
Lineament, that itself was not reactivated during the Mesozoic (Dusar & Langenaeker, 1992;
Langenaeker, 2000). We therefore consider the presence of oblique Carboniferous strike-slip fault
systems as a likely cause for changes in strain distribution between the middle Mesozoic fault
populations, such as the FFS. These differences in strain distributions persisted when the fault
populations were reactivated during the Late Cretaceous and Cenozoic tectonic phases.
**5.3 The role of non-colinear faults in accommodating segmentation**
The southern and northern structural domains of the FFS as identified in this study are both
dominated by NW-SE striking faults. The boundary between these domains is sharp, as it coincides
with the oblique (non-colinear), WNW-ESE striking GBF. The GBF transitions the FFS from localized
faulting in a narrow southern border zone to distributed faulting in a wide norther border zone. The
transition from localized to distributed faulting is not abrupt, but stepwise as strain is redistributed

along the GBF from the Neeroeteren fault at its eastern tip towards connecting faults (of the northern structural domain) further west:

- o As the Neeroeteren fault branches into the GBF and Bocholt faults, the total throw of the Neeroeteren Fault is divided between these two faults (Figs. 7 & 8).
- o At the bifurcation between the GBF and Reppel faults, the total throw of the GBF also decreases by an amount equal to the throw of the Reppel Fault (Fig. 7 & 8).
- o At the location where the GBF dies out, no more large displacement faults were observed in the northern domain (Fig. 7).

The GBF thereby causes a major left-stepping pattern in the FFS. North and south of the study area, other major WNW-ESE striking faults, such as the Veldhoven and Lövenicher faults (see Fig. 1 for their locations), are associated with similar left-stepping patterns and even larger geometrical changes in the border fault systems of the Cenozoic Roer Valley Rift System. At both of their lateral tips, the Veldhoven and Lövenicher-Kast faults connect to large (total syn-rift throws >200 m) NW-SE striking faults (Klett et al., 2002; Vernes et al., 2018). The GBF is only known to be delimited along its eastern fault tip by the major NW-SE striking Neeroeteren fault, while due to lack of data coverage, the geometry of its western fault tip remains uncertain. However, given the limited total syn-rift throw (100 m) at its westernmost seismically covered section, a connection with a major NW-SE fault here seems unlikely. This is consistent with the smaller maximum vertical Cenozoic throw along the GBF (<250 m) compared to the Veldhoven and Lövenicher faults (locally >500 m). Geological maps of the thickness of the Chalk Group in the Netherlands (Duin et al., 2006) illustrate that the Veldhoven fault, just like the GBF, was also already of major importance during the Late Cretaceous inversion of the RVG. The non-colinear faults therefore played an important role in accommodating long-lived strain redistribution along the RVG border fault systems, under phases of both compression and extension.

## 6. Conclusions

The Roer Valley Graben is bounded by large, NW-SE striking border fault systems that probably developed during the middle Mesozoic. During phases of Late Cretaceous contraction and Cenozoic extension, these border fault systems were reactivated. The western border fault system, i.e. the Feldbiss fault system (FFS), is located in northeastern Belgium. Based on careful evaluation of the new geological 3D model of Flanders (northern Belgium), this study shows the presence of two structural domains in the FFS with distinctly different strain distributions during both Late Cretaceous compression and Cenozoic extension:

- The southern domain is characterized by narrow (< 3 km wide) localized faulting, with Late Cretaceous reverse throws of over 200 m and Cenozoic normal throw of almost 600 m.
- The northern domain is characterized by broad (>10 km wide) distributed faulting and tilting of the pre-inversion and pre-rift strata during these subsequent phases.

The total amount of normal and reverse throws in the two domains of the FFS was estimated to be similar during both tectonic phases. This shows that each domain accommodated a similar amount of deformation, but distributed it differently, whether during inversion or extension. This emphasizes that pre-existing structural domains in faults systems can have a strong influence on the later fault reactivation.

Between both structural domains of the FFS, a major NNE-SSW striking latest Carboniferous transpressional structure was recently mapped, called the Gruitrode Lineament. As was illustrated in the East African Rift System, pre-existing lineaments can be the cause of segmentation and redistribution of strain in rift border fault systems. Further southwest and parallel to the Gruitrode Lineament, another latest Carboniferous transpressional structure (Donderslag Lineament) is known to coincide with an important change in fault patterns. The oblique Carboniferous strike-slip fault systems are therefore considered as a likely cause for the changes in strain distribution within the

middle Mesozoic FFS, which persisted as this system was reactivated during the Late Cretaceous and
Cenozoic tectonic phases.
The faults in the two structural domains of the FFS strike dominantly NW-SE, but the change in
geometry between them takes place across the oblique WNW-ESE striking Grote Brogel fault. This
fault thereby progressively widened the FFS in northern direction, redistributing localized strain from
predominantly a single fault in the southern domain into several smaller faults in the northern
domain. Also in other parts of the Roer Valley Graben, WNW-ESE striking faults are associated with
major geometrical changes (left-stepping patterns) in its border fault system.

**Acknowledgements**
We gratefully acknowledge financial support from the Bureau for Environment and Spatial
Development - Flanders. We would like to thank K. van Baelen for her work on the figures. We thank
A. Malz, C. Jackson and J. Kley for their helpful reviews and recommendations that led to
considerable improvements of the manuscript.

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

**Figures**


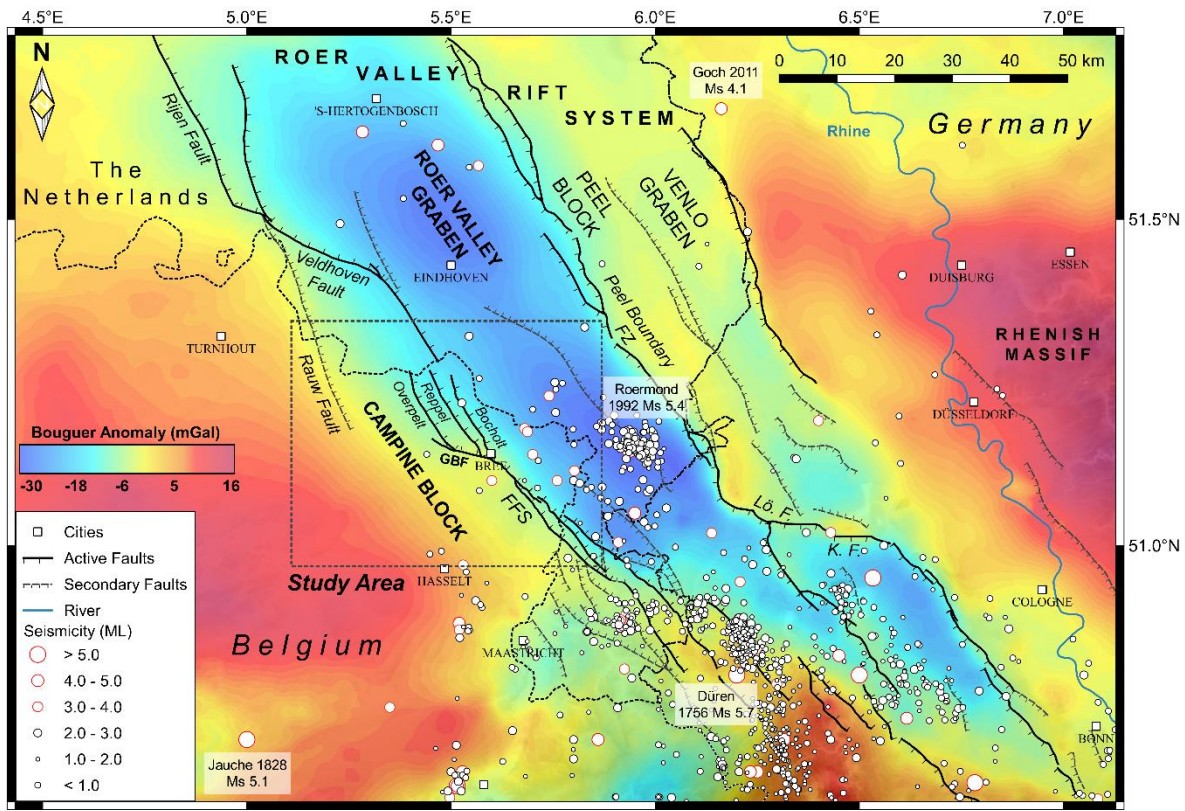

**Fig. 1:** The Roer Valley Rift System with its different tectonic blocks, border fault configuration and
seismicity in relation to the Bouguer anomaly. Note the lower Bouguer Anomaly values (Bouguer
data gathered by the ROB as described in Everaerts & De Vos, 2012, and Verbeurgt et al., 2019) in
the Roer Valley Graben related to the thick Cenozoic sequence of uncompacted sediments. The grey
dashed square indicates the study area and the location of Figs. 3, 4 and 5. Lö. F.: Lövenich Fault; K.
F.: Kast Fault. Surface fault traces modified after Vanneste et al. (2013) and Deckers et al. (2018).
Historical and instrumental natural seismicity updated until Dec. 2019 (ROB catalog, 1350-2019).

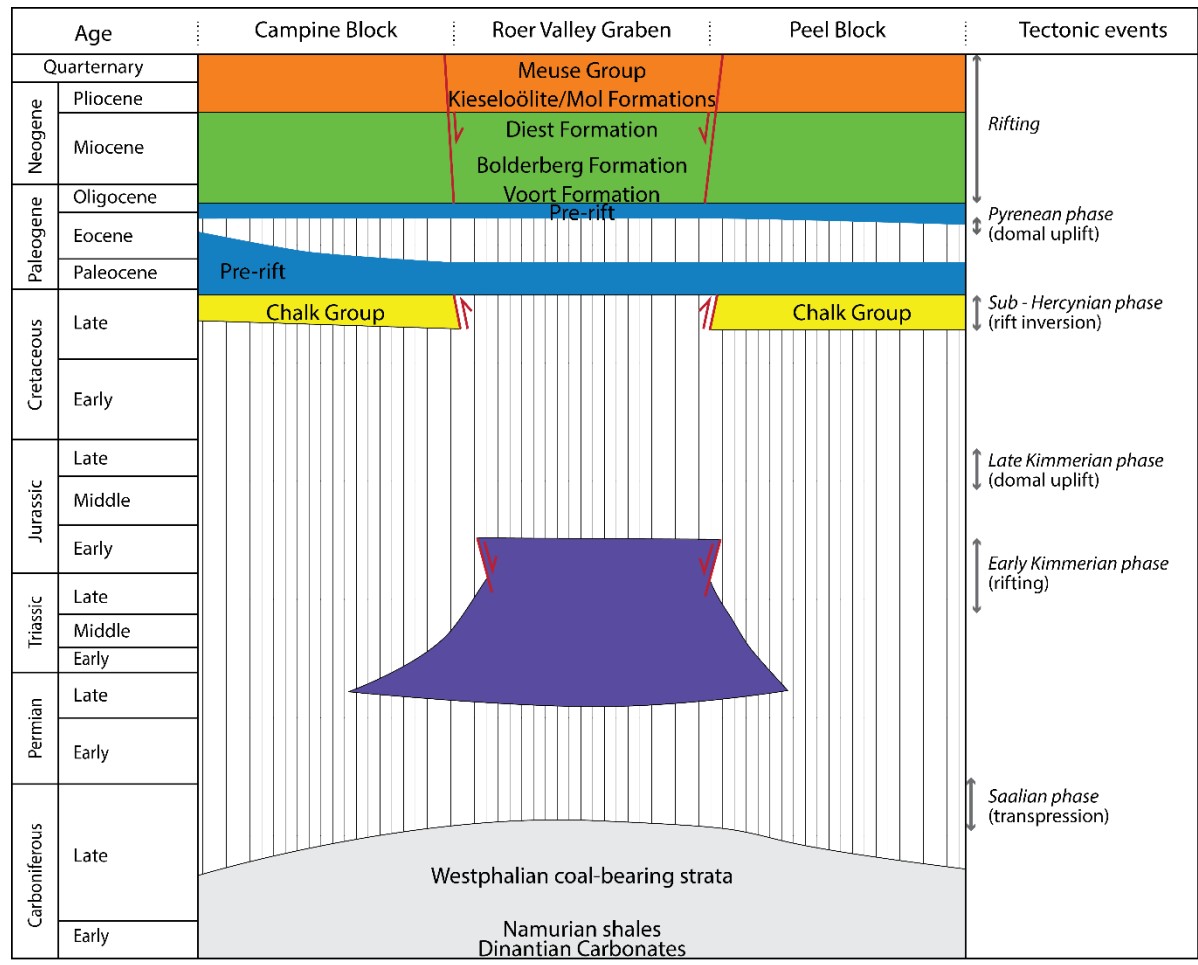

**Fig. 2:** General stratigraphy, ages and the main tectonic phases within the Campine Block, Roer Valley
Graben and Peel Block. Figure modified after Geluk et al. (1994).

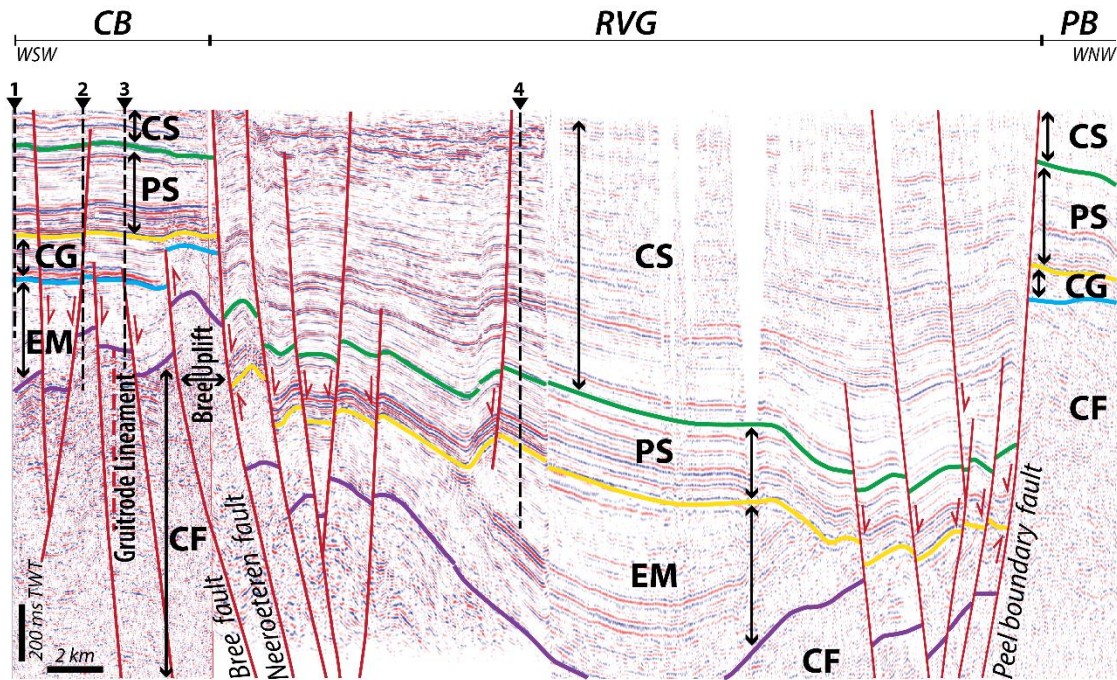

**Fig. 3:** Composite seismic section (constructed from three seismic lines) from the Campine Block (CB)
in the west, across the Roer Valley Graben (RVG) in the centre up to the Peel Block in the east. The
location of this section is shown in Figure 5. Note the presence of thick early to middle Mesozoic
strata, but absence of the Chalk Group within the RVG. The western part of this section extends
across the southern domain of this study, and highlights the intersection with the latest
Carboniferous Gruitrode Lineament. CF= Carboniferous strata; CG= Chalk Group; CS= Cenozoic syn-
rift strata; EM= Early and middle Mesozoic strata; GL= axis of the Gruitrode Lineament; PS= pre-rift
strata. Numbers represent boreholes at or nearby the seismic lines: 1= Meeuwen (DOV-code:
kb18d48w-B173; 2= Gruitrode (DOV-code: kb18d48w-B186; 3= Bree (DOV-code: kb18d48w-B193;
4= Molenbeersel (DOV-code: kb18d49w-B226).

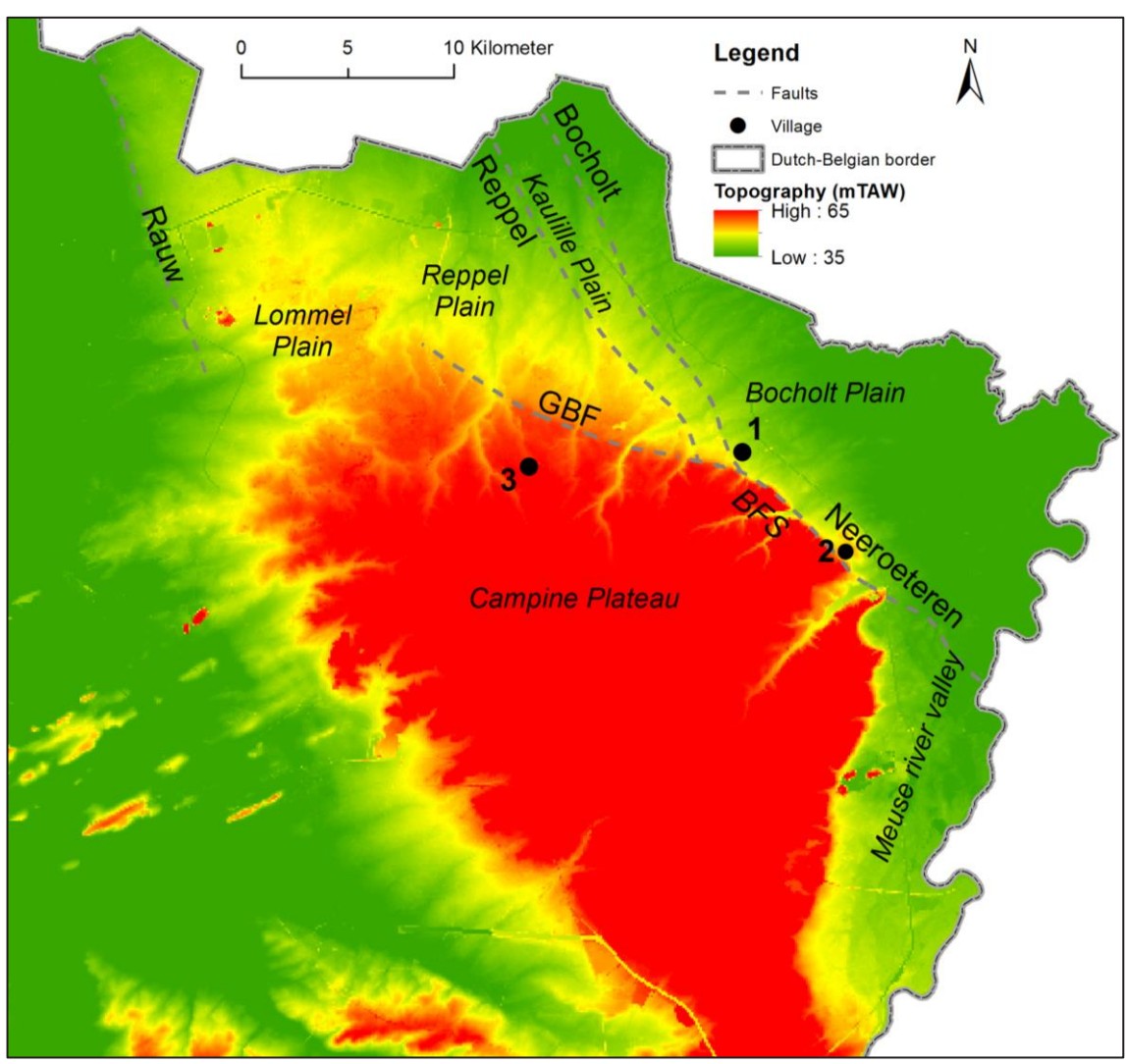

**Fig. 4:** Topography in the study area with indication of the main morphological features and faults of
the G3Dv3-model that have a topographic expression. BFS= Bree Fault Scarp; GBF = Grote Brogel
fault; Numbers denote villages: 1 = Bree; 2 = Waterloos; 3 = Peer. DTMV-II model from Agentschap
Informatie Vlaanderen (2018).

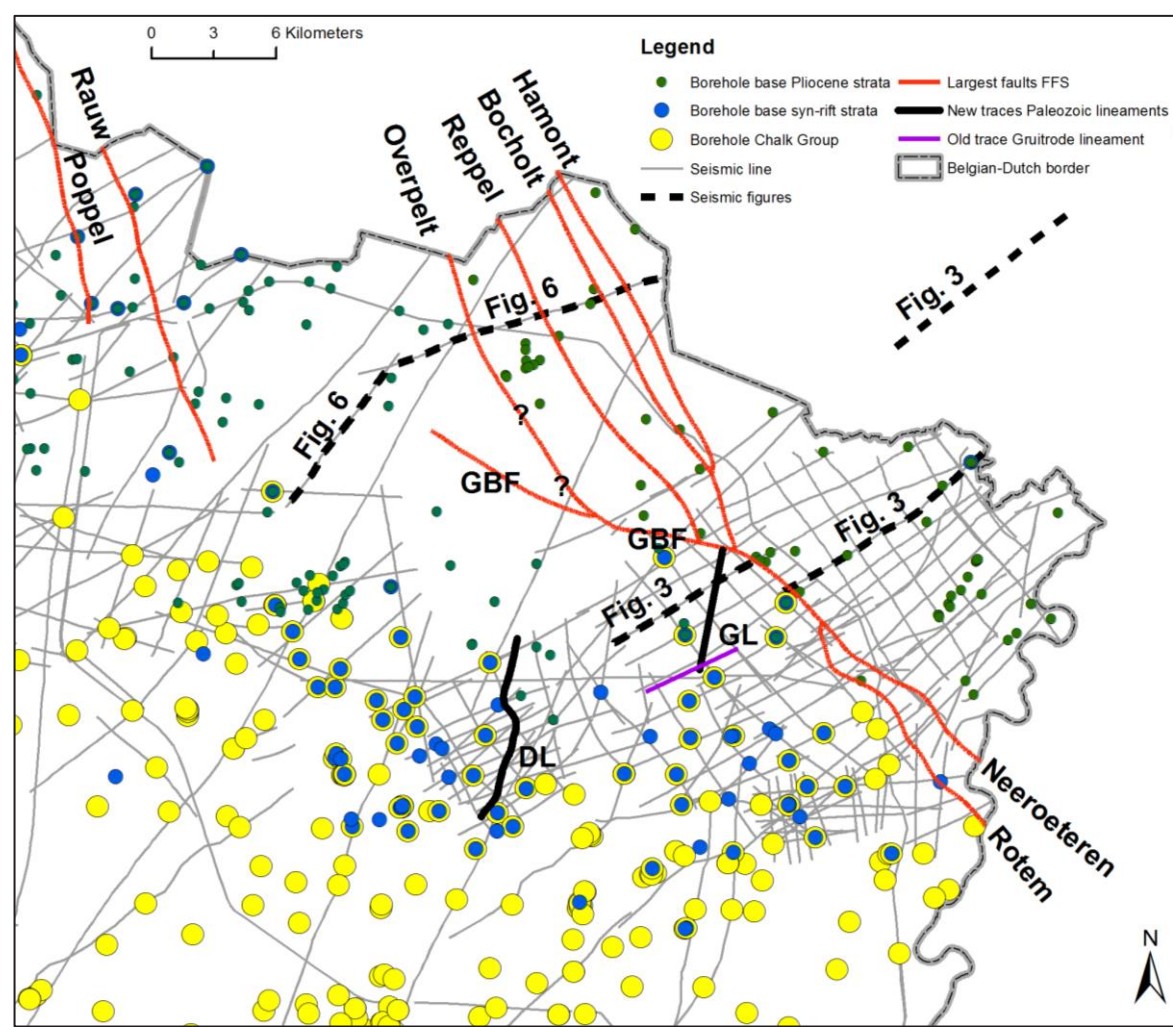

**Fig. 5:** Overview map of the main input-data used for the G3Dv3-model of the area, namely seismic
lines and borehole selections (for mapping of the bases of the Pliocene, syn-rift strata and Chalk
Group). The composite seismic sections of figures 3 and 6 are marked in bold (dashed lines). The
numbers of the boreholes in figure 3 are indicated. The modelled major fault lines of the FFS are
marked by red lines, while the modelled axes of the late Paleozoic Donderslag Lineament (DL) and
Gruitrode Lineament (GL) are marked by black lines. The old trace of the Gruitrode Lineament by
Langenaeker (2000) is shown in purple. Question marks indicate uncertainties in the fault trace.

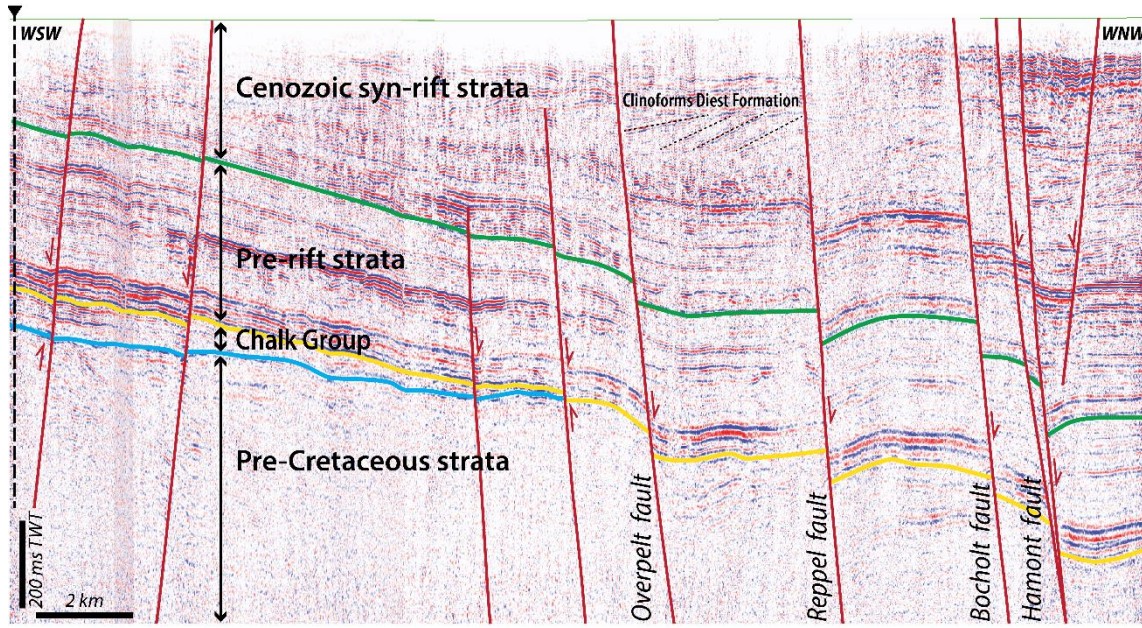

**Fig. 6:** Composite seismic section across the northern structural domain of this study. The location
of this section is shown in Figure 5. Note the gradual decrease in thickness of the Chalk Group and
increase in thickness of the Cenozoic syn-rift strata from west to east along the western part of this
section. In the eastern part of this section, the Chalk Group is absent and Cenozoic syn-rift strata
thicken stepwise across faults. The seismic expressions of some of the westward-prograding
clinoforms in the Upper Miocene Diest Formation are indicated. In the westernmost part of this
section, the deep borehole Lommel (DOV-code: kb17d47w-B262) is indicated.


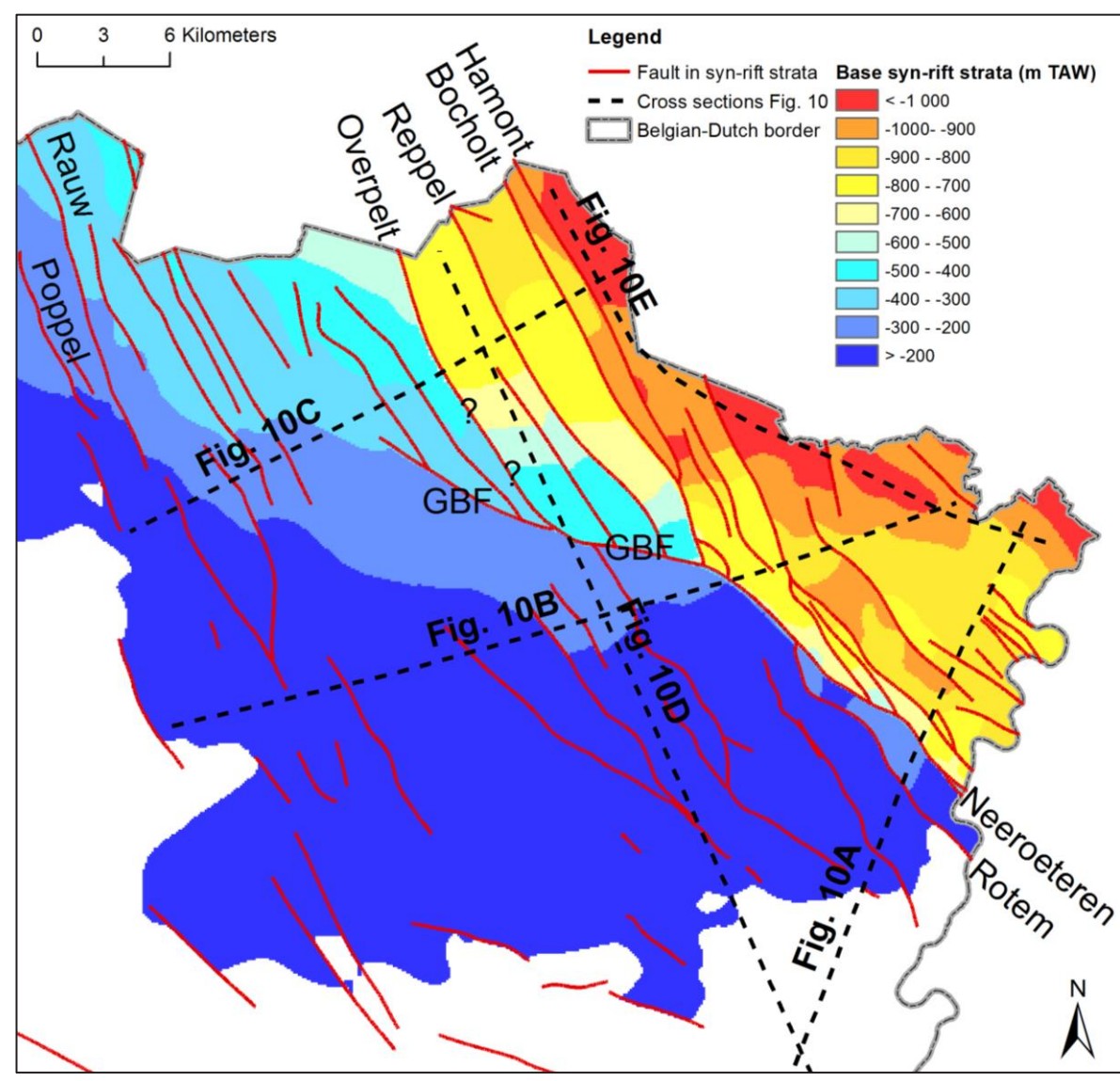

**Fig. 7:** Map showing the depth of the base of the syn-rift strata and the syn-rift faults in the study
area from the G3Dv3-model. The locations of the cross-sections in figure 10 are also indicated.
Question marks indicate uncertainties in the fault trace.


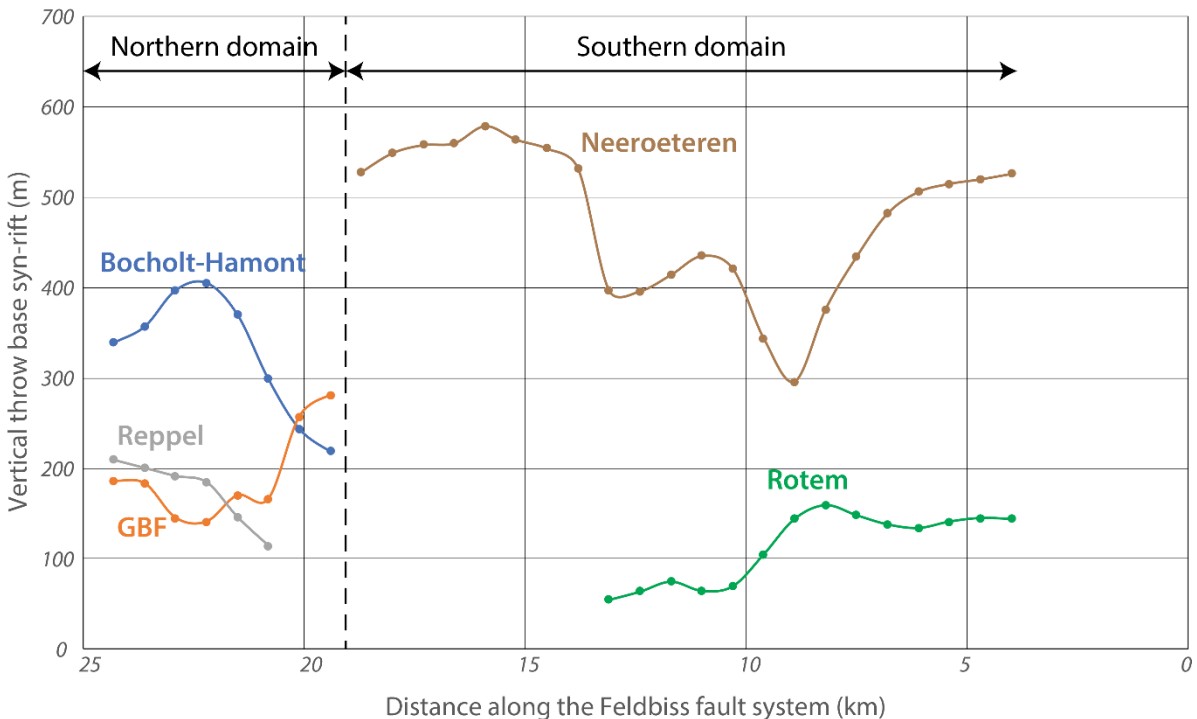


**Fig. 8:** Vertical throws along the major faults of the FFS based on the G3Dv3-model. This trace runs
from the Belgian/Dutch border in the southeast towards the supposed bending of the GBF into the
Overpelt fault in the northwest. Notice the abrupt change in vertical throw of faults at the boundary
between the northern and southern domain.

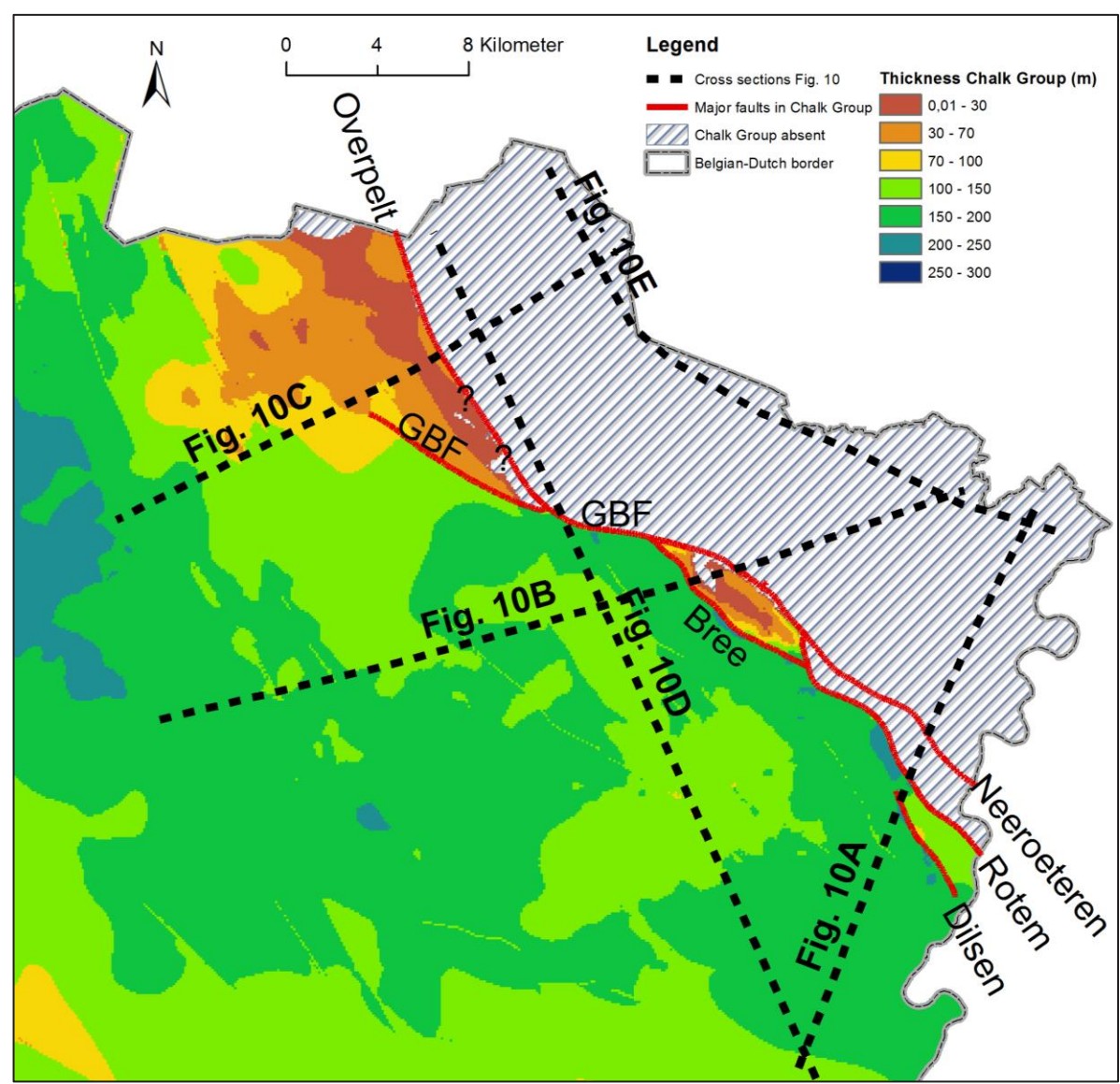

**Fig. 9:** Map showing the thickness of the Chalk Group and major reverse/thrust faults that influenced it in the study area from the G3Dv3-model. The locations of the cross-sections in figure 10 are indicated. Note that our informal definition of the Chalk Group only contains those parts of the formal Chalk Group that were deposited during inversion of the RVG, and are therefore missing in the latter. Younger formations of the formal Chalk Group are grouped in the pre-rift strata for the purpose of this study. Question marks indicate uncertainties in the fault trace.

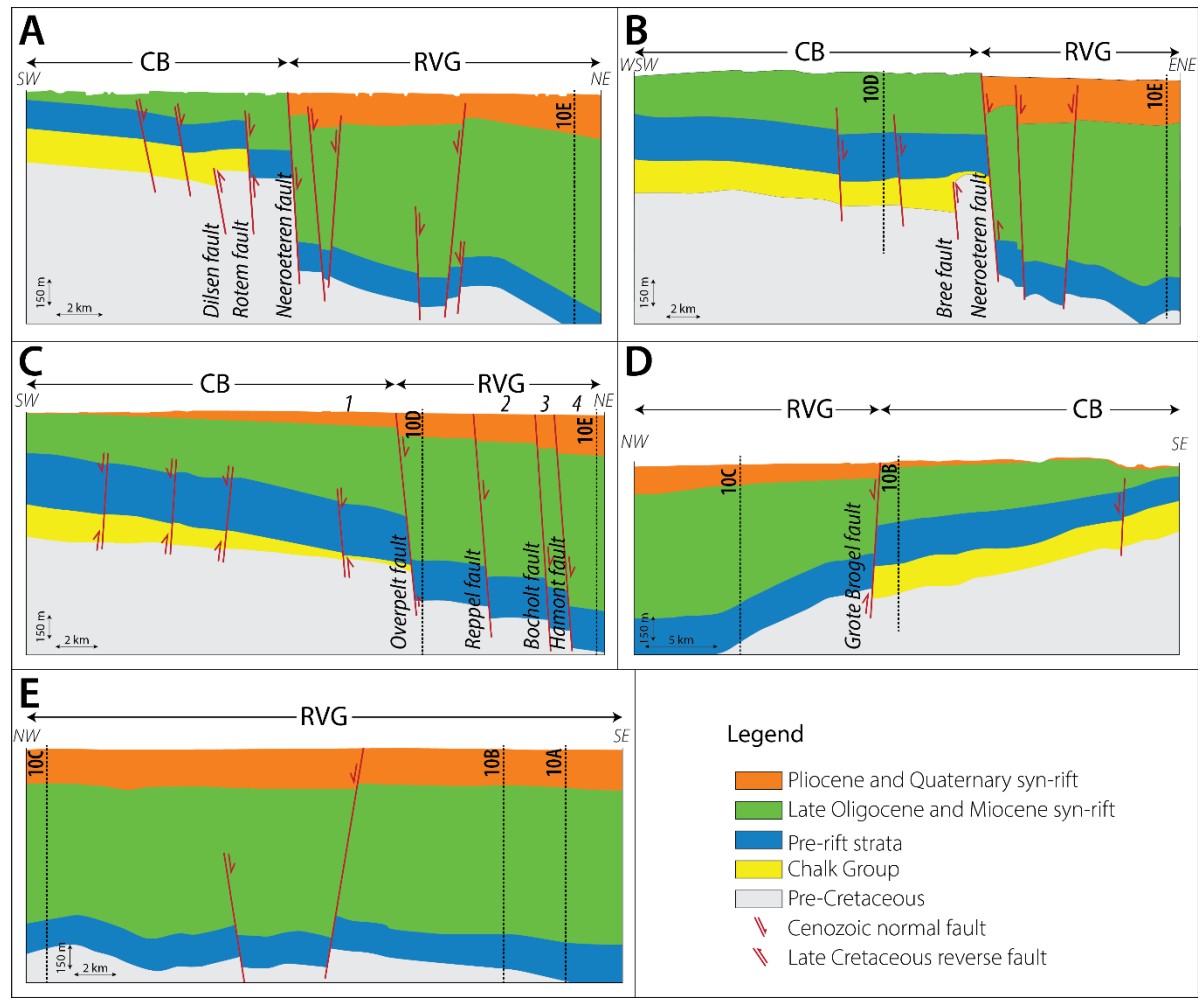

**Fig. 10.** Sections constructed from the G3Dv3-model across (A, B and C) and along (D and E) the FFS. Sections A and B cross the southern domain of the FFS, while section C crosses the northern domain. Section D runs from the footwall of the southern domain of the FFS (or in the CB) into the northern domain of the FFS. Section E runs across the hangingwall of the FFS or in the RVG. The locations of these sections are indicated on figures 7 and 9. 1= Lommel Plain; 2= Reppel Plain; 3= Kaulille Plain; 4= Bocholt Plain.