# Peer review of "Influence of inherited structural domains and their particular strain distributions on the Roer Valley Graben evolution from inversion to extension"

_Solid Earth, 2020_

## Referee Comment (RC1) · Christopher Jackson (Referee) · 14 Apr 2020

Department of Earth Science & Engineering Imperial College Prince Consort Rd London SW7 2BP England UK

Tuesday 14th April

Dear Editor/Authors,

Thanks for giving me the opportunity to review manuscript se-2020-23 ('Influence of inherited structural domains and their particular strain distributions on the Roer Valley Graben evolution from inversion to extension'). The general focus of the paper should

be of great interest to the general readership of Solid Earth, as well as this Special Issue on 'Inversion Tectonics'. The numbers below refer to specific numbers in the manuscript.

1. Rather than starting with the specifics of the study area, the Abstract might benefit from a more general sentence (or two) on the generic issues to be tackled in the paper (e.g. strain partitioning during inversion). By doing this, the paper may more immediately appeal to a broader, more general audience; e.g. the reader may not be particularly expert or interested in the Roer Valley, but may be concerned with the far wider, more general topic of basin inversion. 2. I do not follow this section of text, especially the last sentence in the paragraph; i.e. how does the similar strain distributions show the importance of inherited structural domains? Please be more specific. Also, given the rifting width is narrow in the south than the north, and that the magnitude of extension and contraction was the same between the two domains, does this mean that there were: (i) fewer, larger displacement normal faults; and (ii) a greater amount of reverse reactivation per fault, in the south? 3. It is not clear why segmentation is mentioned at this point in the Abstract. It might work better earlier in the Abstract, when you describe the overall (present) structural style of the study, and before you discuss the kinematics (i.e. before the last few sentences in the first paragraph). 4. The last sentence of the Abstract does not really make any clear statements about the inversion aspect of the study; instead, it principally focuses on rifting. This is surprisingly, given the Special Issue is about inversion tectonics. 5. Like the Abstract, the start of the Introduction is rather focused on NW Europe in general, and the Roer Valley in particular. It might help to make some broader, more generic statements about the repeated reactivation (in extension and contraction) of basin-bounding faults. For example, the rationale-style statements in the last two sentences in the first paragraph of the Introduction might be brought to the start of this section. 6. On L59-61, where you mention "stratigraphic distributions", it might also be worth mentioning "isopachs" (i.e. thickness maps), given this is, I think, what you are referring to. 7. L87-105 – Despite being syn-rift, the uppermost Oligocene to Recent strata appears to be rather widespread

and tabular in the stratigraphic column presented in Fig. 2. Why are these units not more locally developed within the Roer Valley Graben, in a similar way to the Jurassic units? Or are these syn-rift units present on the basin flanks, but substantially thinner and/or punctuated by unconformities related to rift-flank uplift/non-deposition? 8. L109 – you here mention the Paleogene-to-Neogene extension direction, but what was the shortening direction associated with the sub-Hercynian compressional phase? You do not mention this near L68-71 in the preceding paragraph. This is very important, given this will ultimately influence whether and how certain faults were reverse reactivated. 9. L110-128 – This text would greatly benefit if some structure maps (e.g. Fig. 5) and/or cross-sections (Figs 7 and 8) were cited. It is presently very difficult to visualise the described relationships in the absence of any graphical support. I sense many such maps and sections have been generated as part of previous studies (e.g. Decker et al., 2019) and have been included in earlier publications, but some of them may benefit from being included again here. For example, a regional, NE-trending cross-section would a very useful accompaniment to Fig. 1. 10. L158-169 - I think it is important to show at least some seismic profiles. If you do not, then the reader has to solely rely on the geoseismic (i.e. interpreted) sections presented in Fig. 7; in my view, this is not sufficient to really convince the reader of your structural and stratigraphic (i.e. thickness) descriptions, and subsequent interpretations and conclusions. I again argue that, although some of these raw data may have been presented in earlier papers, they need showing again here, especially to help the reader visualise some of the interpretation challenges mentioned in, for example, L213-225. 11. L182-196 – Related to comment (10), this section would benefit from one or two simple stratigraphic correlations (e.g. one from the southern and one from the northern domain), perhaps presented next to or below spatially coincident seismic profiles, showing how the main syn-inversion and syn-rift strata change in thickness across some of the key structures. The gridded data in Fig. 6 are useful, as are the cross-sections in Fig. 7, but some hard-data, in the form of a correlation with stratigraphic/formation tops clearly indicated, would strongly support the inversion-to-extension argument being presented. 12. As a general aside,

I recommend the authors read Freeman et al. (2010) - Using empirical geological rules to reduce structural uncertainty in seismic interpretation of faults. J. Struct. Geol. 32, 1668-1676. This is an excellent paper, showing how simple displacement-length (D-L) plots and displacement 'strike-projections' can be used to help reduce interpretation uncertainty in areas of modest-quality (and quantity) seismic reflection data. In any case, the authors are to be congratulated on a very open, honest discussion of the uncertainty in their structural model. 13. L248-251 – I think this text needs modifying, given that structure maps alone say nothing about kinematics; i.e. they tell you about present-day basin structure, but not about the motion history (i.e. kinematics) of those structures. Kinematics are best revealed by isopach maps, stratigraphic correlations (see comment (11)), cross-sections (e.g. Fig. 7), etc, both of which tell you about the timing of structural movement as recorded in the related uplift and subsidence. 14. L263-269 (and elsewhere) – I suggest you use the cross-sections in Fig. 7 to help support your structural descriptions. 15. As stated above, the cross-sections in Fig. 7 would greatly benefit from the addition of the location of boreholes. This would help make the interpretations more convincing; i.e. at the moment, the reader of this particular paper has to simply trust that the geometries, depths, etc, presented in Fig. 7 are true, without supporting data. 16. The title of sub-section 4.1 may benefit from modifying, given you provide a description of the present, rift-related structural style (e.g. fault throw, spacing, length, etc), but not the kinematics of rifting. 17. L346 – change "doesn't" to "does not". 18. L345-348 – The fact that strain is the same but more diffuse in the northern domain than the southern domain is a very important, which is currently described in a rather qualitative manner. One option would be to actually quantify this relationship by measuring and summing throws (or heaves) along a series of broadly fault-normal (i.e. NE-trending) profiles in the northern and southern domains (e.g. Wilson, P., Elliott, G.M., Gawthorpe, R.L., Jackson, C.A-L., Michelsen, L., & Sharp, I.R. (2013). Geometry and segmentation of an evaporite-detached normal fault array: 3D seismic analysis of the southern Bremstein Fault Complex, offshore mid-Norway. Journal of Structural Geology, 51, 74-91). This would be a powerful

addition to the paper, and make the segmentation argument, which is currently only really supported by three cross-sections, even more compelling. 19. L352-355 - What data indicate that the Chalk Group is missing in the axis of the Roer Valley Graben? It is shown as being absent in the cross-sections in Fig. 7, but is this directly constrained by deep boreholes in this location? For example, is the Cenozoic pre-rift early in demonstrably direct contact with Pre-Cretaceous strata in the rift axis? This query relates back to my earlier suggestion that far more stratigraphic and seismic data need to be shown to support Fig. 7. 20. L359-360 – A key issue relates to the argument that only few faults presently have reverse throws. From what I can see in Fig. 7, all faults are still in net-extension; even the Bree, Dilsen and Rotem faults, for example, all appear to be in net-extension, despite reverse slip vector arrows being drawn at deeper depths. So I again ask, "what data are constraining the interpretations presented in Fig. 7?". Furthermore, classic inversion-related structures, like so-called 'harpoon structures' (i.e. hangingwall anticlines) are absent. As it stands, I see little solid evidence for inversion in the data as it is currently presented. 21. L378-382 – I do not follow the argument that the Bree and Dilsen faults formed only during the sub-Hercynian inversion event, and are not pre-existing, rift-related normal faults that were subsequently inverted (although see my comment (20) regarding the present lack of evidence for inversion). Why do you think this is the case? It is completely implausible that they are Late Jurassic structures? 22. L392-393 – I cannot see reverse throw of 200 m at the stratigraphic level of the Chalk Group in Fig. 7B. The GBF appears to be in net-extension along its entire dip extent. This comment also applies to the start of section 5 (Discussion and Conclusion), where you argue for the magnitude of reverse throw along the various faults in the northern and southern domains. 23. L414-415 - Related to comment (22), this is a critical statement, which is presently not strongly supported by the presented data. I also strongly recommend the authors read Reilly et al. (2017) - https://sp.lyellcollection.org/content/439/1/447, who come to a similar conclusion, but via the presentation of much more quantitative data. 24. L415 – Mora et al. (2008) is not in the reference list. Please check all references. 25. L426-428

– Why would footwall shortcut faults be less prone to being reactivation (in extension) than other faults? Is it because they have gentler dips, thus are essentially 'locked' due to the normal stress exceeded the imposed (extensional) shear stress? 26. L447-454 – I agree there is a change in structural style between the two domains across the GBF, but why does this happen? More specifically, why are more fault required in the northern domain than the southern domain to accommodate the same extensional strain?

In summary, this is an very interesting piece of work that, as stated above, will be of interest to the readership of the 'Inversion Tectonics' Special Issue of Solid Earth. I am keen to see these data published, given the lack of case studies explicitly focused on the role of fault size in controlling the structural style and partitioning of inversion tectonics. However, as I hope is clear from my comments above, I believe additional work and modifications will help improve the paper. For example, more focus on the fault size issue, as shown so clearly by Reilly et al. (2017), would make this a very strong contribution. In general, the English and grammar are good; there are, however, many places where these could be improved. Note that I started to edit a hard-copy of the manuscript, but this was taking a considerable amount of time; I thus encourage the authors (and editorial office) to very closely read future versions of the manuscript.

I am more-than-happy for the authors to contact me to discuss any of the issues raised in my review.

Yours sincerely

Christopher Jackson (c.jackson@imperial.ac.uk)

---

## Referee Comment (RC2) · Alexander Malz (Referee) · 22 Apr 2020

Dear authors and editors,

first of all, I would like to thank you for the interesting manuscript (se-2020-23), which handels with the influence of inherited structures to the reactivation potential of the Roer Valley Graben. The focus of the manuscript is of great interest to the readership of Solid Earth, particularly for that special issue on inversion tectonics. It is in general well-written and easy to read. The authors provide several illustrations that help the reader to understand the descriptions. Nevertheless, there are several issues to improve and probably make the manuscript more interesting for a broader audience. 1.

[Figure]

As it stands now the manuscript is more or less a local study of the RVG, which is interesting and important as well. Clarifying what we can learn from your study applicable to similar tectonic settings would highly enhance the importance of your manuscript. In the introduction you should give a short (one or two sentences) overview of the 'state of the art' in our understanding of inversion tectonics. What are the main controlling factors (mechanical weakening, fault orientation, strain distribution, thermal heating, etc.) for or typical structural features associated with inverted systems. You can then pick up these e.g. factors in your conclusions, which will provide a nice 'frame' for your observation/study. 2. A more or less general or regional overview is missing. Adding an overview map with the location of your study area would be nice. Furthermore, I was wondering about a missing illustration of e.g. a subcrop map of the base Cenozoic and a cross-section including pre-Chalk Group strata, which would help to better understand the situation. 3. As a general comment: All regional and local names used in the text should be shown in at least one figure. Keep in mind that your audience is not familar with local geographical and geological names. 4. Even if the authors concentrate on the extensional reactivation of the RVG, the RVG initially formed as a graben during Jurassic times and became contractionally overprinted during the Cretaceous. During the Cenozoic the RVG became reactivated again under extension. How are the relations - and thus the reactivation potential - between Jurassic, in parts reactivated normal faults and Cretaceous reverse faults (and footwall shortcuts)? Which faults became extensionally reactivated? Is there any relationship between mechanical weakening due to repeated fault activity or between the geometry and kinematics (fault dip or initial sense of slip) and their reactivation potential during extension? 5. The "Dataset und methodology" chapter needs some improvements. As it stands know, it is still unclear for me whether the authors made (1) a new model presented in that study, (2) made the G3Dv3-model for that study or (3) extracted parts or maps and cross-sections from the G3Dv3-model. Either in the abstract "In Flanders, a new geological model was created..." or in chapter 3.3 "...we constructed a map view..." some misleading information is provided. Therefore I strongly suggest reworking the methods

chapter to clarify what was done for exactly that study. Which software was used? Did you generate a 3D-model or a "map-based" GIS-model, etc.? 6. In chapter 4 (results) the authors should think about chapter captions and the associated text. In general, the text gives a very extensive description of individual structural features. In contrast, captions suggest that processes leading to these structures are described. Probably restructuring this chapter a little bit, would improve the manuscript. Therefor, I suggest to separate the 'results' chapter into 'description of model results' (very concise) and 'interpretation and indications for fault kinematics'. 7. Separate the chapter 5. Discuss your interpretation and afterwards precisely write your conclusion. Do not mix! 8. The polyphase evolution of the RVG make some descriptions difficult to follow. Especially in chapter 4 there are plenty descriptions of fault throws and the authors should carefully check their description. E.g. (L359-360) "Due to Cenozoic normal reactivation, only few faults in the study area have net reverse throws as the result of Late Cretaceous compression." If that is the case, how can we ensure that these faults were reactivated? Is there any indication for past fault throws on single faults? How is throw distributed on single faults (e.g. for the pre-Chalk Group strata, syn-inversion strata and rift-strata)? Do the model have the potential to show throw distribution on single faults and for single horizons (e.g. by use of Allan Maps, etc.)? If yes, that would significantly help to illustrate and understand strain distribution across the RVG. 9. Furthermore, there are some detailed comments to the text: a. L41: What means "both" here? You mention at least three stratigraphic units. b. L56: Please specify the used data. Is it reflection or refraction seismics? What kind of borehole data was used? c. L58: What means "basement" in that context? Crystalline or sedimentary "sub-décollement" strata? I suggest to clarify/define that. d. L70: Here, you mention "Chalk Group". I highly suggest to provide ages and chronostratigrapic names. Please keep in mind that most of your readers are not familar with the local stratigraphic names of the RVG region. e. L77-78: Although I understand the intension of this from the modeller's point of view. Nevertheless, it seems a little bit confusing that Mesozoic strata is named 'Cenozoic'. Probably, it would be helpful to modify/enhance the stratigraphic overview figure (Fig.

2) and include some 'real' stratigraphic horizons in relation to your 'model stratigraphy'. f. L87: Please use (or define) the names of structures. What is the Roer Valley Rift System? The Roer Valley Graben? Or is the graben a part of that system? You should check the entire manuscript and use unique names. g. L91: "main faults or those with the largest displacement" - Especially in inverted systems with a high potential to fault reactivation this definition is problematic. Delete that part or provide a definition for 'main faults'. h. L92: Please check the consistent use of abbreviations. The use of 'CB vs. Peel Block vs. RVG' in one sentence isn't good style. i. L94: What is the 'Oligocene Voort Formation'? That should be shown in your stratigraphic chart. j. L329-330: Contradictory numbers (150m vs. 100m)?

As said above, I strongly suggest the publication of this work. Even if this is still one local piece, such case studies will significantly improve our understanding of inversion tectonics. Furthermore, the study shows how geological modelling can help to understand even complex structures like the RVG and their kinematics. If systematically interpreted and evaluated this third - or, if analysed for various chronostratigraphical horizons as done in that study, fourth - dimension enable various new insights into 'inversion tectonics'. Some modifications and additional work will significantly improve the manuscript.

Congratulations for that very interesting contribution. Kind regards, A. Malz

---

## Author Comment (AC1) · 15 Jun 2020

1. Rather than starting with the specifics of the study area, the Abstract might benefit from a more general sentence (or two) on the generic issues to be tackled in the paper (e.g. strain partitioning during inversion). By doing this, the paper may more immediately appeal to a broader, more general audience; e.g. the reader may not be particularly expert or interested in the Roer Valley, but may be concerned with the far wider, more general topic of basin inversion.

We have added several sentences to describe the more generic issues of our manuscript.

[Figure]

2. I do not follow this section of text, especially the last sentence in the paragraph; i.e. how does the similar strain distributions show the importance of inherited structural domains? Please be more specific. Also, given the rifting width is narrow in the south than the north, and that the magnitude of extension and contraction was the same between the two domains, does this mean that there were: (i) fewer, larger displacement normal faults; and (ii) a greater amount of reverse reactivation per fault, in the south?

We have changed this sentence into the following: "The total normal and reverse throws in the two domains of the FFS were estimated to be similar during both tectonic phases. This shows that each domain accommodated a similar amount of compressional and extensional deformation, but persistently distributed it differently." Indeed, there are fewer, larger displacement faults in the southern domain and more in the northern domain. This is mentioned as follows: "A southern domain is characterized by narrow (< 3 km) localized faulting, while the northern is characterized by wide (>10 km) distributed faulting".

3. It is not clear why segmentation is mentioned at this point in the Abstract. It might work better earlier in the Abstract, when you describe the overall (present) structural style of the study, and before you discuss the kinematics (i.e. before the last few sentences in the first paragraph).

We have now mentioned segmentation at the first few sentences of the abstract.

4. The last sentence of the Abstract does not really make any clear statements about the inversion aspect of the study; instead, it principally focuses on rifting. This is surprisingly, given the Special Issue is about inversion tectonics.

The last sentence was removed since it is now already covered by the new first few sentences of the abstract.

5. Like the Abstract, the start of the Introduction is rather focused on NW Europe in general, and the Roer Valley in particular. It might help to make some broader,

more generic statements about the repeated reactivation (in extension and contraction) of basin-bounding faults. For example, the rationale-style statements in the last two sentences in the first paragraph of the Introduction might be brought to the start of this section.

We have now added several sentences to broaden the scope before focusing on the study area. "Rift basins are typically bounded by large fault systems. These border fault systems are generally segmented along strike. As they represent zones of pre-existing weaknesses, the large border fault systems are prone to reactivation under either extension or compression. The effects of pre-existing segmentation upon extensional or compressional strain distributions in reactivated rift border fault systems have thus far received little attention. One of the ideal areas to study these effects is at the border fault systems of the Roer Valley Graben (RVG). These systems developed in the middle Mesozoic, and were reversely reactivated under Late Cretaceous contraction and experienced normal reactivation again under Cenozoic extension (Demyttenaere, 1989; Geluk et al., 1994)."

6. On L59-61, where you mention "stratigraphic distributions", it might also be worth mentioning "isopachs" (i.e. thickness maps), given this is, I think, what you are referring to.

Indeed, we have added the "thickness maps".

7. L87-105 – Despite being syn-rift, the uppermost Oligocene to Recent strata appears to be rather widespread and tabular in the stratigraphic column presented in Fig. 2. Why are these units not more locally developed within the Roer Valley Graben, in a similar way to the Jurassic units? Or are these syn-rift units present on the basin flanks, but substantially thinner and/or punctuated by unconformities related to rift-flank uplift/non-deposition?

The syn-rift strata are much thinner on the graben flanks. This is mentioned in the sentence: "As a result of continuous rifting since the late Oligocene, the abovementioned stratigraphic units are relatively thick in the RVG (over 1000 m) compared to the flanking CB and Peel Blocks (generally below 500 m; Demyttenaere, 1989; Geluk, 1990; Fig. 3)." This is now also shown on the composite seismic figure 3 and on the cross-sections of figure 10.

8. L109 – you here mention the Paleogene-to-Neogene extension direction, but what was the shortening direction associated with the sub-Hercynian compressional phase? You do not mention this near L68-71 in the preceding paragraph. This is very important, given this will ultimately influence whether and how certain faults were reverse reactivated.

We have added the following sentence: "Inversion in the area probably took place under a N-S to NNW-SSE direction of maximum horizontal compression (de Jager, 2003) as the result of convergence between Africa and Europe (Kley and Voigt, 2008)."

9. L110-128 – This text would greatly benefit if some structure maps (e.g. Fig. 5) and/or cross-sections (Figs 7 and 8) were cited. It is presently very difficult to visualise the described relationships in the absence of any graphical support. I sense many such maps and sections have been generated as part of previous studies (e.g. Decker et al., 2019) and have been included in earlier publications, but some of them may benefit from being included again here. For example, a regional, NE-trending cross-section would a very useful accompaniment to Fig. 1.

We have added one composite seismic section on figure 3 that runs across the RVG (and the southern structural domain of the FFS). This figure shows the stratigraphic distributions and provides support for the 3D-models mentioned in the text. We have also added a composite seismic section as Figure 6 that runs across the northern structural domain of the FFS.

10. L158-169 - I think it is important to show at least some seismic profiles. If you do not, then the reader has to solely rely on the geoseismic (i.e. interpreted) sections presented in Fig. 7; in my view, this is not sufficient to really convince the reader of

your structural and stratigraphic (i.e. thickness) descriptions, and subsequent interpretations and conclusions. I again argue that, although some of these raw data may have been presented in earlier papers, they need showing again here, especially to help the reader visualise some of the interpretation challenges mentioned in, for example, L213-225.

We have added two composite seismic section as figures 3 & 6 in order to convince the reader of the differentiation between the northern and southern structural domains of this study.

11. L182-196 – Related to comment (10), this section would benefit from one or two simple stratigraphic correlations (e.g.one from the southern and one from the northern domain), perhaps presented next to or below spatially coincident seismic profiles, showing how the main syn-inversion and syn-rift strata change in thickness across some of the key structures. The gridded data in Fig. 6 are useful, as are the cross-sections in Fig. 7, but some hard-data, in the form of a correlation with stratigraphic/formation tops clearly indicated, would strongly support the inversion-to-extension argument being presented.

We have indeed added one composite seismic section across the northern domain and one across the southern domain (figures 3 & 6). Several of the used, deep boreholes are also indicated.

12. As a general aside, I recommend the authors read Freeman et al. (2010) - Using empirical geological rules to reduce structural uncertainty in seismic interpretation of faults. J. Struct. Geol. 32, 1668-1676. This is an excellent paper, showing how simple displacement-length (D-L) plots and displacement 'strike-projections' can be used to help reduce interpretation uncertainty in areas of modest-quality (and quantity) seismic reflection data. In any case, the authors are to be congratulated on a very open, honest discussion of the uncertainty in their structural model.

This is indeed a very interesting paper to handle uncertainties and improve fault interpretations on 2D seismic data. We will keep that in mind for the next modelling campagnes.

13. L248-251 – I think this text needs modifying, given that structure maps alone say nothing about kinematics; i.e. they tell you about present-day basin structure, but not about the motion history (i.e. kinematics) of those structures. Kinematics are best revealed by isopach maps, stratigraphic correlations (see comment (11)), cross-sections (e.g. Fig. 7), etc, both of which tell you about the timing of structural movement as recorded in the related uplift and subsidence.

We changed "kinematics" into "geometry".

14. L263-269 (and elsewhere) – I suggest you use the cross-sections in Fig. 7 to help support your structural descriptions.

We have added some additional references to the figures.

15. As stated above, the cross-sections in Fig. 7 would greatly benefit from the addition of the location of boreholes. This would help make the interpretations more convincing; i.e. at the moment, the reader of this particular paper has to simply trust that the geometries, depths, etc, presented in Fig. 7 are true, without supporting data.

We now show some of the important, deep boreholes on the composite seismic sections of figures 3 and 6.

16. The title of sub-section 4.1 may benefit from modifying, given you provide a description of the present, rift-related structural style (e.g. fault throw, spacing, length, etc), but not the kinematics of rifting. 17

We agree and have changed the title into: " Structural style of Cenozoic rifting"

17. L346 – change "doesn't" to "does not".

This sentence was modified accordingly.

18. L345-348 – The fact that strain is the same but more diffuse in the northern domain than the southern domain is a very important, which is currently described in a rather qualitative manner. One option would be to actually quantify this relationship by measuring and summing throws (or heaves) along a series of broadly fault-normal (i.e. NE-trending) profiles in the northern and southern domains (e.g. Wilson, P., Elliott, G.M., Gawthorpe, R.L., Jackson, C.A-L., Michelsen, L., & Sharp, I.R. (2013). Geometry and segmentation of an evaporite-detached normal fault array: 3D seismic analysis of the southern Bremstein Fault Complex, offshore mid-Norway. Journal of Structural Geology, 51, 74-91). This would be a powerful addition to the paper, and make the segmentation argument, which is currently only really supported by three cross-sections, even more compelling.

We agree and have added an additional figure 8 to the manuscript with the Cenozoic throw distribution along some of the major faults in the southern and northern domains.

19. L352-355 – What data indicate that the Chalk Group is missing in the axis of the Roer Valley Graben? It is shown as being absent in the cross-sections in Fig. 7, but is this directly constrained by deep boreholes in this location? For example, is the Cenozoic pre-rift early in demonstrably direct contact with Pre-Cretaceous strata in the rift axis? This query relates back to my earlier suggestion that far more stratigraphic and seismic data need to be shown to support Fig. 7.

The supporting evidence by deep boreholes is now shown on figure 3. The Molenbeersel borehole only penetrates the youngest sequences of the formal Chalk Group which are – for the purpose of this manuscript – not included in the Chalk Group (see chapter 2.1 of geological setting and stratigraphy).

20. L359-360 – A key issue relates to the argument that only few faults presently have reverse throws. From what I can see in Fig. 7, all faults are still in net-extension; even the Bree, Dilsen and Rotem faults, for example, all appear to be in net-extension, despite reverse slip vector arrows being drawn at deeper depths. So I again ask,

"what data are constraining the interpretations presented in Fig. 7?". Furthermore, classic inversion-related structures, like so-called 'harpoon structures' (i.e. hangingwall anticlines) are absent. As it stands, I see little solid evidence for inversion in the data as it is currently presented.

The Bree and Dilsen faults were not normally reactivated and display net reverse throws on figure 10 (former figure 7). The net reverse throw of the Bree fault is supported by figure 3.

21. L378-382 – I do not follow the argument that the Bree and Dilsen faults formed only during the sub-Hercynian inversionn event, and are not pre-existing, rift-related normal faults that were subsequently inverted (although see my comment (20) regarding the present lack of evidence for inversion). Why do you think this is the case? It is completely implausible that they are Late Jurassic structures?

We cannot exclude earlier activity, but find it very unlikely. We therefore have rewritten this section as follows: "If they indeed represent footwall shortcut faults, the Bree and Dilsen faults would have originated during Late Cretaceous compression to accommodate inversion on the pre-existing Neeroeteren and Rotem faults. This hypothesis is supported by the fact that the base of the Lower to Middle Mesozoic strata shows a very similar amount of reverse vertical throw as the base of the Chalk Group along the Bree fault (Fig. 3). Nevertheless, earlier (Cimmerian) activity along the Bree and Dilsen fault cannot be excluded. Contrary to most other faults in the FFS, the Bree and Dilsen faults were not reactivated during Cenozoic extension and therefore now still have net reverse throws (Figs. 3, 10A and -B)."

22. L392-393 – I cannot see reverse throw of 200 m at the stratigraphic level of the Chalk Group in Fig. 7B. The GBF appears to be in net-extension along its entire dip extent. This comment also applies to the start of section 5 (Discussion and Conclusion), where you argue for the magnitude of reverse throw along the various faults in the northern and southern domains.

We have now clarified this statement by rewriting it into: "The Chalk Group is about 200 m thick in the footwall of the GBF (the CB), but absent in its hangingwall (the northern domain of the FFS), which indicates that this fault had a reverse throw of at least 200 m (Fig. 10D)".

23. L414-415 - Related to comment (22), this is a critical statement, which is presently not strongly supported by the presented data. I also strongly recommend the authors read Reilly et al. (2017) - https://sp.lyellcollection.org/content/439/1/447, who come to a similar conclusion, but via the presentation of much more quantitative data.

This statement is supported by the maps of figures 7 and 9, as well as the cross-sections of figure 10. Thanks for the article of Reilly et al. (2017)! It is indeed very relevant also for this manuscript, so we have added a reference to it in paragraph 5.1.

24. L415 – Mora et al. (2008) is not in the reference list. Please check all references.

We have corrected it to Mora et al. (2009) which is included in the reference.

25. L426-428 – Why would footwall shortcut faults be less prone to being reactivation (in extension) than other faults? Is it because they have gentler dips, thus are essentially 'locked' due to the normal stress exceeded the imposed (extensional) shear stress?

Good question. Since it is not the focus of our manuscript, we simply wrote in the text that we presume that the middle Mesozoic normal faults, rather than the Late Cretaceous footwall shortcut faults, were preferential sites for Cenozoic normal reactivation.

26. L447-454 – I agree there is a change in structural style between the two domains across the GBF, but why does this happen? More specifically, why are more fault required in the northern domain than the southern domain to accommodate the same extensional strain?

We consider underlying changes in lithospheric strength as likely triggers for the differences in strain distribution. This is now discussed in paragraph "5.2 Possible mechanisms behind the segmentation".

---

## Author Comment (AC2) · 15 Jun 2020

1. As it stands now the manuscript is more or less a local study of the RVG, which is interesting and important as well. Clarifying what we can learn from your study applicable to similar tectonic settings would highly enhance the importance of your manuscript. In the introduction you should give a short (one or two sentences) overview of the 'state of the art' in our understanding of inversion tectonics. What are the main controlling factors (mechanical weakening, fault orientation, strain distribution, thermal heating, etc.) for or typical structural features associated with inverted systems. You can then pick up these e.g. factors in your conclusions, which will provide a nice 'frame'

for your observation/study.

We have added several sentences to describe the more generic issues of our manuscript. By doing so, we have also better described the issue we are dealing with: the influence of pre-existing graben border fault segmentation on strain-distribution during reactivation.

2. A more or less general or regional overview is missing. Adding an overview map with the location of your study area would be nice. Furthermore, I was wondering about a missing illustration of e.g. a subcrop map of the base Cenozoic and a cross-section including pre-Chalk Group strata, which would help to better understand the situation.

We have added a figure 3 of composite seismic lines across the RVG to illustrate the regional overview. It shows the major stratigraphic units (Carboniferous, middle Mesozoic, Upper Cretaceous, pre-rift and syn-rift) and structural features (Gruitrode Lineament, CB, RVG, Peel Block).

3. As a general comment: All regional and local names used in the text should be shown in at least one figure. Keep in mind that your audience is not familar with local geographical and geological names.

We agree and have done so.

4. Even if the authors concentrate on the extensional reactivation of the RVG, the RVG initially formed as a graben during Jurassic times and became contractionally overprinted during the Cretaceous. During the Cenozoic the RVG became reactivated again under extension. How are the relations - and thus the reactivation potential - between Jurassic, in parts reactivated normal faults and Cretaceous reverse faults (and footwall shortcuts)? Which faults became extensionally reactivated? Is there any relationship between mechanical weakening due to repeated fault activity or between the geometry and kinematics (fault dip or initial sense of slip) and their reactivation potential during extension?

[Figure]

We have now provided a better discussion on the fault evolution from the middle Meso-zoic up to the Cenozoic in chapter "5.1 Graben border activity and segmentation".

5. The "Dataset und methodology" chapter needs some improvements. As it stands know, it is still unclear for me whether the authors made (1) a new model presented in that study, (2) made the G3Dv3-model for that study or (3) extracted parts or maps and crosssections from the G3Dv3-model. Either in the abstract "In Flanders, a new geological model was created: : :" or in chapter 3.3 ": : :we constructed a map view: : :" some misleading information is provided. Therefore I strongly suggest reworking the methodschapter to clarify what was done for exactly that study. Which software was used? Did you generate a 3D-model or a "map-based" GIS-model, etc.?

We have modified the chapter Dataset and Methodology accordingly by adding sen-tences that we extracted maps and cross-section from the G3Dv3-model by ArcGIS and iMOD software.

6. In chapter 4 (results) the authors should think about chapter captions and the asso-ciated text. In general, the text gives a very extensive description of individual structural features. In contrast, captions suggest that processes leading to these structures are described. Probably restructuring this chapter a little bit, would improve the manuscript. Therefore, I suggest to separate the 'results' chapter into 'description of model results' (very concise) and 'interpretation and indications for fault kinematics'.

We have changed the captions of chapter 4. The former caption "Late Oligocene to recent rifting" was changed into "Structural style of Cenozoic rifting", while the former caption "Late Cretaceous compression" was changed into "Structural style of Late Cre-taceous inversion".

7. Separate the chapter 5. Discuss your interpretation and afterwards precisely write your conclusion. Do not mix!

We have added a chapter 6. Conclusions

8. The polyphase evolution of the RVG make some descriptions difficult to follow. Especially in chapter 4 there are plenty descriptions of fault throws and the authors should carefully check their description. E.g. (L359-360) "Due to Cenozoic normal reactivation, only few faults in the study area have net reverse throws as the result of Late Cretaceous compression." If that is the case, how can we ensure that these faults were reactivated? Is there any indication for past fault throws on single faults? How is throw distributed on single faults (e.g. for the pre-Chalk Group strata, syn-inversion strata and rift-strata)? Do the model have the potential to show throw distribution on single faults and for single horizons (e.g. by use of Allan Maps, etc.)? If yes, that would significantly help to illustrate and understand strain distribution across the RVG.

We use the thickness changes across faults of the Late Cretaceous Chalk Group and Cenozoic syn-rift strata to evaluate their syn-compression and syn-rift throws, respectively. In order to better visualize throw distribution along the FFS, we have added figure 8 that shows the vertical throw of the base of the syn-rift strata along the major faults in the FFS in both northern and southern structural domains.

9. Furthermore, there are some detailed comments to the text:

a. L41: What means "both" here? You mention at least three stratigraphic units. We have changed this sentence into :"This is indicated by gravimetric maps of the area (Fig. 1) and in more detail in geological maps of the middle Mesozoic (Jurassic), Upper Cretaceous and Cenozoic stratigraphic distributions and thicknesses in the area (c.f. Duin et al., 2006; Deckers et al., 2019)."

b. L56: Please specify the used data. Is it reflection or refraction seismics? What kind of borehole data was used?

We have changed this sentence into: "The G3Dv3-model of the area was created by the integration and interpretation of all available 2D seismic reflection and borehole data (borehole descriptions and wireline logs)."

c. L58: What means "basement" in that context? Crystalline or sedimentary "sub-décollement" strata? I suggest to clarify/define that.

We have changed the word "basement" into "strata", since the term basement is not relevant for this study.

d. L70: Here, you mention "Chalk Group". I highly suggest to provide ages and chronostratigrapic names. Please keep in mind that most of your readers are not familar with the local stratigraphic names of the RVG region.

We have added the chronostratigraphic names Campanian to middle Maastrichtian.

e. L77-78: Although I understand the intension of this from the modeller's point of view. Nevertheless, it seems a little bit confusing that Mesozoic strata is named 'Cenozoic'. Probably, it would be helpful to modify/enhance the stratigraphic overview figure (Fig. 2) and include some 'real' stratigraphic horizons in relation to your 'model stratigraphy'.

We have removed the "Cenozoic" from the "Cenozoic pre-rift strata" and now only refer to them as pre-rift strata. This is also mentioned in the text as follows: "For the purpose of this study, the latest Maastrichtian to early Oligocene strata are here referred to as the pre-rift strata". We have now also included real stratigraphic names for the syn-rift strata (Voort, Bolderberg, Diest, etc. formations).

f. L87: Please use (or define) the names of structures. What is the Roer Valley Rift System? The Roer Valley Graben? Or is the graben a part of that system? You should check the entire manuscript and use unique names.

These structures are defined in the text of chapter 2.1 as follows: "Major fault activity resumed in the late Oligocene, when the Roer Valley Rift System developed as a northwest-trending branch of the Rhine-Graben-System (Ziegler, 1988), throughout the south-eastern part of the Netherlands, eastern Flanders and adjacent parts of Germany (Fig. 1). This system currently extends over a distance of roughly 200 km and has a width of up to 75 km. Those faults with the strongest displacements divide the

central Roer Valley Rift System into the Campine Block in the west, the pre-existing Roer Valley Graben in the center and Peel Block in the east."

These different structural features (now also including the Roer Valley Rift System) are also shown in figure 1.

g. L91: "main faults or those with the largest displacement" - Especially in inverted systems with a high potential to fault reactivation this definition is problematic. Delete that part or provide a definition for 'main faults'.

We have deleted the word "main".

h. L92: Please check the consistent use of abbreviations. The use of 'CB vs. Peel Block vs. RVG' in one sentence isn't good style.

We have changed the abbreviations in this sentence into actual names, so CB to Campine Block and RVG to Roer Valley Graben

i. L94: What is the Oligocene Voort Formation'? That should be shown in your stratigraphic chart.

We have added this stratigraphic unit to the stratigraphic chart of figure 2.

j. L329-330: Contradictory numbers (150m vs. 100m)?

The vertical throw along the GBF decreases from 250 m towards 150 m, or by 100 m.

As said above, I strongly suggest the publication of this work. Even if this is still one local piece, such case studies will significantly improve our understanding of inversion tectonics. Furthermore, the study shows how geological modelling can help to understand even complex structures like the RVG and their kinematics. If systematically interpreted and evaluated this third - or, if analysed for various chronostratigraphical horizons as done in that study, fourth - dimension enable various new insights into 'inversion tectonics'. Some modifications and additional work will significantly improve the manuscript. Congratulations for that very interesting contribution. Kind regards, A.

Malz

Please also note the supplement to this comment:
https://se.copernicus.org/preprints/se-2020-23/se-2020-23-AC2-supplement.pdf

**Supplement:**

Dear Editor,

Many thanks for sending us the thorough and careful remarks by the reviewers. Their annotations improved significantly our manuscript.

In accordance with the remarks by the reviewers, some of the major textual changes of the manuscript are:

- **Abstract:** We added some additional sentences to describe the more generic issues of our manuscript. In the last sentences of the abstract, we now also discuss some new insights based on a recent modelling campaign of the Upper Carboniferous strata by Rombaut et al. (2020). They modelled a NNE-SSW striking major strike-slip zone that crosses the junction between the southern and northern domains of the FFS. This type of major strike-slip faults, as well as the distinction of the FFS into two separate domains with their particular strain distributions, may both be related to underlying changes in the lithospheric strengths.

- **1 Introduction**: We added some additional sentences to describe the more generic issues of our manuscript and also made some textual improvements.

- **2 Geological setting and stratigraphy**: We added information on the geological setting and stratigraphy of the Upper Paleozoic strata, since they are now relevant as explanation for the segmentation of the FFS (latest Carboniferous strike-slip faults cross the junction between the northern and southern domains). Some additional information was given on the driving mechanism and orientation of maximum horizontal stress during late Cretaceous compression, next to several textual changes.

- **3. Dataset and methodology:** In the paragraph **3.2 Dataset in the study area**, a paragraph was added on the Palaeozoic lineaments, since they are now relevant for this article. In paragraph **3.3 Methodology**, the tools/software for the reconstruction of the figures were now added (ArcGIS, iMOD).

- **4. Results:** The titles of paragraphs **4.1** and **4.2** were changed to give a better description of their content. Also several textual changes were made to make the text more comprehensible.

- **5. Discussion:** This chapter was now subdivided into three paragraphs:
  The first paragraph discusses the general segmentation of the FFS and insights that they provide on the importance of pre-existing segmentation on later reactivation. This paragraph now provides better understanding on the existing knowledge and new observations made by this study on segmentation of the FFS.
  The second paragraph discusses the possible mechanisms behind segmentation. This now also includes the very recent work by Rombaut et al. (2020) who observed a major strike-slip fault/lineament at the junction between the southern and northern domains. The presence of this strike-slip fault might provide the clue on why segmentation of the FFS took place at this location.
  The third paragraph discusses the role of non-colinear faults in accommodating segmentation. In this paragraph, the comparison to other non-colinear fault systems in the Roer Valley Graben was added.

- **6. Conclusions:** This was newly added.

For the figures, the following changes were made:

- We added two composite seismic lines across the study area: now called figures 3 and 6. Figure 3 crosses the southern domain of the FFS, the entire Roer Valley Graben and shows the expression of the latest Carboniferous Gruitrode Lineament. Figure 6 crosses the northern domain of the FFS.
- Figures 1 and 4 (older figure 3) remained the same
- Figure 2 was extended for the Carboniferous interval and now also includes some stratigraphic names.
- Figure 5 now also shows the locations of the composite seismic sections and latest Carboniferous Donderslag and Gruitrode Lineaments
- Figure 7 and 9 remained more or less the same.
- Figure 8 is new and represents the Cenozoic throw along some of the major faults in the FFS from the southern domain into the northern domain.
- Figure 10 now integrates the previously separated cross-sections (former figure 7). One additional cross-section (10E) was added, which is located along the hangingwall of the FFS.

Below, you can find the responses (in red) to the individual remarks/questions of the reviewers (in blue):

**Review by Christopher Jackson**

Dear Editor/Authors,
Thanks for giving me the opportunity to review manuscript se-2020-23 ('Influence of inherited structural domains and their particular strain distributions on the Roer Valley Graben evolution from inversion to extension'). The general focus of the paper should be of great interest to the general readership of Solid Earth, as well as this Special Issue on 'Inversion Tectonics'. I have written numerous comments directly on the manuscript, which I have scanned and attached to this review. The numbers below refer to specific numbers in the manuscript.

1. Rather than starting with the specifics of the study area, the Abstract might benefit from a more general sentence (or two) on the generic issues to be tackled in the paper (e.g. strain partitioning during inversion). By doing this, the paper may more immediately appeal to a broader, more general audience; e.g. the reader may not be particularly expert or interested in the Roer Valley, but may be concerned with the far wider, more general topic of basin inversion.

*We have added several sentences to describe the more generic issues of our manuscript.*

2. I do not follow this section of text, especially the last sentence in the paragraph; i.e. how does the similar strain distributions show the importance of inherited structural domains? Please be more specific. Also, given the rifting width is narrow in the south than the north, and that the magnitude of extension and contraction was the same between the two domains, does this mean that there were: (i) fewer, larger displacement normal faults; and (ii) a greater amount of reverse reactivation per fault, in the south?

*We have changed this sentence into the following:*

*"The total normal and reverse throws in the two domains of the FFS were estimated to be similar during both tectonic phases. This shows that each domain accommodated a similar amount of compressional and extensional deformation, but persistently distributed it differently."*

*Indeed, there are fewer, larger displacement faults in the southern domain and more in the northern domain. This is mentioned as follows: "A southern domain is characterized by narrow (< 3 km) localized faulting, while the northern is characterized by wide (>10 km) distributed faulting".*

3. It is not clear why segmentation is mentioned at this point in the Abstract. It might work better earlier in the Abstract, when you describe the overall (present) structural style of the study, and before you discuss the kinematics (i.e. before the last few sentences in the first paragraph).

*We have now mentioned segmentation at the first few sentences of the abstract.*

4. The last sentence of the Abstract does not really make any clear statements about the inversion aspect of the study; instead, it principally focuses on rifting. This is surprisingly, given the Special Issue is about inversion tectonics.

*The last sentence was removed since it is now already covered by the new first few sentences of the abstract.*

5. Like the Abstract, the start of the Introduction is rather focused on NW Europe in general, and the Roer Valley in particular. It might help to make some broader, more generic statements about the repeated reactivation (in extension and contraction) of basin-bounding faults. For example, the rationale-style statements in the last two sentences in the first paragraph of the Introduction might be brought to the start of this section.

We have now added several sentences to broaden the scope before focusing on the study area. *"Rift basins are typically bounded by large fault systems. These border fault systems are generally segmented along strike. As they represent zones of pre-existing weaknesses, the large border fault systems are prone to reactivation under either extension or compression. The effects of pre-existing segmentation upon extensional or compressional strain distributions in reactivated rift border fault systems have thus far received little attention. One of the ideal areas to study these effects is at the border fault systems of the Roer Valley Graben (RVG). These systems developed in the middle Mesozoic, and were reversely reactivated under Late Cretaceous contraction and experienced normal reactivation again under Cenozoic extension (Demyttenaere, 1989; Geluk et al., 1994)."*

6. On L59-61, where you mention "stratigraphic distributions", it might also be worth mentioning "isopachs" (i.e. thickness maps), given this is, I think, what you are referring to.

*Indeed, we have added the "thickness maps".*

7. L87-105 – Despite being syn-rift, the uppermost Oligocene to Recent strata appears to be rather widespread and tabular in the stratigraphic column presented in Fig. 2. Why are these units not more locally developed within the Roer Valley Graben, in a similar way to the Jurassic units? Or are these syn-rift units present on the basin flanks, but substantially thinner and/or punctuated by unconformities related to rift-flank uplift/non-deposition?

The syn-rift strata are much thinner on the graben flanks. This is mentioned in the sentence: *"As a result of continuous rifting since the late Oligocene, the abovementioned stratigraphic units are relatively thick in the RVG (over 1000 m) compared to the flanking CB and Peel Blocks (generally below 500 m; Demyttenaere, 1989; Geluk, 1990; Fig. 3)."* This is now also shown on the composite seismic figure 3 and on the cross-sections of figure 10.

8. L109 – you here mention the Paleogene-to-Neogene extension direction, but what was the shortening direction associated with the sub-Hercynian compressional phase? You do not mention this near L68-71 in the preceding paragraph. This is very important, given this will ultimately influence whether and how certain faults were reverse reactivated.

We have added the following sentence: *"Inversion in the area probably took place under a N-S to NNW-SSE direction of maximum horizontal compression (de Jager, 2003) as the result of convergence between Africa and Europe (Kley and Voigt, 2008)."*

9. L110-128 – This text would greatly benefit if some structure maps (e.g. Fig. 5) and/or cross-sections (Figs 7 and 8) were cited. It is presently very difficult to visualise the described relationships in the absence of any graphical support. I sense many such maps and sections have been generated as part of previous studies (e.g. Decker et al., 2019) and have been included in earlier publications, but some of them may benefit from being included again here. For example, a regional, NE-trending cross-section would a very useful accompaniment to Fig. 1.

*We have added one composite seismic section on figure 3 that runs across the RVG (and the southern structural domain of the FFS). This figure shows the stratigraphic distributions and provides support for the 3D-models mentioned in the text. We have also added a composite seismic section as Figure 6 that runs across the northern structural domain of the FFS.*

10. L158-169 - I think it is important to show at least some seismic profiles. If you do not, then the reader has to solely rely on the geoseismic (i.e. interpreted) sections presented in Fig. 7; in my view, this is not sufficient to really convince the reader of your structural and stratigraphic (i.e. thickness) descriptions, and subsequent interpretations and conclusions. I again argue that, although some of these raw data may have been presented in earlier papers, they need showing again here, especially to help the reader visualise some of the interpretation challenges mentioned in, for example, L213-225.

*We have added two composite seismic section as figures 3 & 6 in order to convince the reader of the differentiation between the northern and southern structural domains of this study.*

11. L182-196 – Related to comment (10), this section would benefit from one or two simple stratigraphic correlations (e.g.one from the southern and one from the northern domain), perhaps presented next to or below spatially coincident seismic profiles, showing how the main syn-inversion and syn-rift strata change in thickness across some of the key structures. The gridded data in Fig. 6 are useful, as are the cross-sections in Fig. 7, but some hard-data, in the form of a correlation with stratigraphic/formation tops clearly indicated, would strongly support the inversion-to-extension argument being presented.

*We have indeed added one composite seismic section across the northern domain and one across the southern domain (figures 3 & 6). Several of the used, deep boreholes are also indicated.*

12. As a general aside,
I recommend the authors read Freeman et al. (2010) - Using empirical geological rules to reduce structural uncertainty in seismic interpretation of faults. J. Struct. Geol. 32, 1668-1676. This is an excellent paper, showing how simple displacement-length (D-L) plots and displacement 'strike-projections' can be used to help reduce interpretation uncertainty in areas of modest-quality (and quantity) seismic reflection data. In any case, the authors are to be congratulated on a very open, honest discussion of the uncertainty in their structural model.

*This is indeed a very interesting paper to handle uncertainties and improve fault interpretations on 2D seismic data. We will keep that in mind for the next modelling campagnes.*

13. L248-251 – I think this text needs modifying, given that structure maps alone say nothing about kinematics; i.e. they tell you about present-day basin structure, but not about the motion history (i.e. kinematics) of those structures. Kinematics are best revealed by isopach maps, stratigraphic correlations (see comment (11)), cross-sections (e.g. Fig. 7), etc, both of which tell you about the timing of structural movement as recorded in the related uplift and subsidence.

*We changed "kinematics" into "geometry".*

14. L263-269 (and elsewhere) – I suggest you use the cross-sections in Fig. 7 to help support your structural descriptions.

*We have added some additional references to the figures.*

15. As stated above, the cross-sections in Fig. 7 would greatly benefit from the addition of the location of boreholes. This would help make the interpretations more convincing; i.e. at the moment, the reader of this particular paper has to simply trust that the geometries, depths, etc, presented in Fig. 7 are true, without supporting data.

*We now show some of the important, deep boreholes on the composite seismic sections of figures 3 and 6.*

16. The title of sub-section 4.1 may benefit from modifying, given you provide a description of the present, rift-related structural style (e.g. fault throw, spacing, length, etc), but not the kinematics of rifting. 17

*We agree and have changed the title into: " Structural style of Cenozoic rifting"*

17. L346 – change "doesn't" to "does not".

*This sentence was modified accordingly.*

18. L345-348 – The fact that strain is the same but more diffuse in the northern domain than the southern domain is a very important, which is currently described in a rather qualitative manner. One option would be to actually quantify this relationship by measuring and summing throws (or heaves)

along a series of broadly fault-normal (i.e. NE-trending) profiles in the northern and southern domains (e.g. Wilson, P., Elliott, G.M., Gawthorpe, R.L., Jackson, C.A-L., Michelsen, L., & Sharp, I.R. (2013). Geometry and segmentation of an evaporite-detached normal fault array: 3D seismic analysis of the southern Bremstein Fault Complex, offshore mid-Norway. Journal of Structural Geology, 51, 74-91). This would be a powerful addition to the paper, and make the segmentation argument, which is currently only really supported by three cross-sections, even more compelling.

*We agree and have added an additional figure 8 to the manuscript with the Cenozoic throw distribution along some of the major faults in the southern and northern domains.*

19. L352-355 – What data indicate that the Chalk Group is missing in the axis of the Roer Valley Graben? It is shown as being absent in the cross-sections in Fig. 7, but is this directly constrained by deep boreholes in this location? For example, is the Cenozoic pre-rift early in demonstrably direct contact with Pre-Cretaceous strata in the rift axis? This query relates back to my earlier suggestion that far more stratigraphic and seismic data need to be shown to support Fig. 7.

*The supporting evidence by deep boreholes is now shown on figure 3. The Molenbeersel borehole only penetrates the youngest sequences of the formal Chalk Group which are – for the purpose of this manuscript – not included in the Chalk Group (see chapter 2.1 of geological setting and stratigraphy).*

20. L359-360 – A key issue relates to the argument that only few faults presently have reverse throws. From what I can see in Fig. 7, all faults are still in net-extension; even the Bree, Dilsen and Rotem faults, for example, all appear to be in net-extension, despite reverse slip vector arrows being drawn at deeper depths. So I again ask, "what data are constraining the interpretations presented in Fig. 7?". Furthermore, classic inversion-related structures, like so-called 'harpoon structures' (i.e. hangingwall anticlines) are absent. As it stands, I see little solid evidence for inversion in the data as it is currently presented.

*The Bree and Dilsen faults were not normally reactivated and display net reverse throws on figure 10 (former figure 7). The net reverse throw of the Bree fault is supported by figure 3.*

21. L378-382 – I do not follow the argument that the Bree and Dilsen faults formed only during the sub-Hercynian inversionn event, and are not pre-existing, rift-related normal faults that were subsequently inverted (although see my comment (20) regarding the present lack of evidence for inversion). Why do you think this is the case? It is completely implausible that they are Late Jurassic structures?

We cannot exclude earlier activity, but find it very unlikely. We therefore have rewritten this section as follows:

*"If they indeed represent footwall shortcut faults, the Bree and Dilsen faults would have originated during Late Cretaceous compression to accommodate inversion on the pre-existing Neeroeteren and Rotem faults. This hypothesis is supported by the fact that the base of the Lower to Middle Mesozoic strata shows a very similar amount of reverse vertical throw as the base of the Chalk Group along the Bree fault (Fig. 3). Nevertheless, earlier (Cimmerian) activity along the Bree and Dilsen fault cannot be excluded. Contrary to most other faults in the FFS, the Bree and Dilsen faults were not reactivated during Cenozoic extension and therefore now still have net reverse throws (Figs. 3, 10A and -B)."*

22. L392-393 – I cannot see reverse throw of 200 m at the stratigraphic level of the Chalk Group in Fig. 7B. The GBF appears to be in net-extension along its entire dip extent. This comment also applies to the start of section 5 (Discussion and Conclusion), where you argue for the magnitude of reverse throw along the various faults in the northern and southern domains.

We have now clarified this statement by rewriting it into:
*"The Chalk Group is about 200 m thick in the footwall of the GBF (the CB), but absent in its hangingwall (the northern domain of the FFS), which indicates that this fault had a reverse throw of at least 200 m (Fig. 10D)".*

23. L414-415 - Related to comment (22), this is a critical statement, which is presently not strongly supported by the presented data. I also strongly recommend the authors read Reilly et al. (2017) - https://sp.lyellcollection.org/content/439/1/447, who come to a similar conclusion, but via the presentation of much more quantitative data.

*This statement is supported by the maps of figures 7 and 9, as well as the cross-sections of figure 10. Thanks for the article of Reilly et al. (2017)! It is indeed very relevant also for this manuscript, so we have added a reference to it in paragraph 5.1.*

24. L415 – Mora et al. (2008) is not in the reference list. Please check all references.

*We have corrected it to Mora et al. (2009) which is included in the reference.*

25. L426-428 – Why would footwall shortcut faults be less prone to being reactivation (in extension) than other faults? Is it because they have gentler dips, thus are essentially 'locked' due to the normal stress exceeded the imposed (extensional) shear stress?

*Good question. Since it is not the focus of our manuscript, we simply wrote in the text that we presume that the middle Mesozoic normal faults, rather than the Late Cretaceous footwall shortcut faults, were preferential sites for Cenozoic normal reactivation.*

26. L447-454 – I agree there is a change in structural style between the two domains across the GBF, but why does this happen? More specifically, why are more fault required in the northern domain than the southern domain to accommodate the same extensional strain?

*We consider underlying changes in lithospheric strength as likely triggers for the differences in strain distribution. This is now discussed in paragraph "5.2 Possible mechanisms behind the segmentation".*

Dear authors and editors,
first of all, I would like to thank you for the interesting manuscript (se-2020-23), which handels with the influence of inherited structures to the reactivation potential of the Roer Valley Graben. The focus of the manuscript is of great interest to the readership of Solid Earth, particularly for that special issue on inversion tectonics. It is in general well-written and easy to read. The authors provide several illustrations that help the reader to understand the descriptions. Nevertheless, there are several issues to improve and probably make the manuscript more interesting for a broader audience.

1. As it stands now the manuscript is more or less a local study of the RVG, which is interesting and important as well. Clarifying what we can learn from your study applicable to similar tectonic settings would highly enhance the importance of your manuscript. In the introduction you should give a short (one or two sentences) overview of the 'state of the art' in our understanding of inversion tectonics. What are the main controlling factors (mechanical weakening, fault orientation, strain distribution, thermal heating, etc.) for or typical structural features associated with inverted systems. You can then pick up these e.g. factors in your conclusions, which will provide a nice 'frame' for your observation/study.

*We have added several sentences to describe the more generic issues of our manuscript. By doing so, we have also better described the issue we are dealing with: the influence of pre-existing graben border fault segmentation on strain-distribution during reactivation.*

2. A more or less general or regional overview is missing. Adding an overview map with the location of your study area would be nice. Furthermore, I was wondering about a missing illustration of e.g. a subcrop map of the base Cenozoic and a cross-section including pre-Chalk Group strata, which would help to better understand the situation.

*We have added a figure 3 of composite seismic lines across the RVG to illustrate the regional overview. It shows the major stratigraphic units (Carboniferous, middle Mesozoic, Upper Cretaceous, pre-rift and syn-rift) and structural features (Gruitrode Lineament, CB, RVG, Peel Block).*

3. As a general comment: All regional and local names used in the text should be shown in at least one figure. Keep in mind that your audience is not familar with local geographical and geological names.

*We agree and have done so.*

4. Even if the authors concentrate on the extensional reactivation of the RVG, the RVG initially formed as a graben during Jurassic times and became contractionally overprinted during the Cretaceous. During the Cenozoic the RVG became reactivated again under extension. How are the relations - and thus the reactivation potential - between Jurassic, in parts reactivated normal faults and Cretaceous reverse faults (and footwall shortcuts)? Which faults became extensionally reactivated? Is there any relationship between mechanical weakening due to repeated fault activity or between the geometry and kinematics (fault dip or initial sense of slip) and their reactivation potential during extension?

*We have now provided a better discussion on the fault evolution from the middle Mesozoic up to the Cenozoic in chapter "5.1 Graben border activity and segmentation".*

5. The "Dataset und methodology" chapter needs some improvements. As it stands know, it is still unclear for me whether the authors made (1) a new model presented in that study, (2) made the G3Dv3-model for that study or (3) extracted parts or maps and crosssections from the G3Dv3-model. Either in the abstract "In Flanders, a new geological model was created: : :" or in chapter 3.3 ": : :we constructed a map view: : :" some misleading information is provided. Therefore I strongly suggest reworking the methodschapter to clarify what was done for exactly that study. Which software was used? Did you generate a 3D-model or a "map-based" GIS-model, etc.?

*We have modified the chapter Dataset and Methodology accordingly by adding sentences that we extracted maps and cross-section from the G3Dv3-model by ArcGIS and iMOD software.*

6. In chapter 4 (results) the authors should think about chapter captions and the associated text. In general, the text gives a very extensive description of individual structural features. In contrast, captions suggest that processes leading to these structures are described. Probably restructuring this chapter a little bit, would improve the manuscript. Therefore, I suggest to separate the 'results' chapter into 'description of model results' (very concise) and 'interpretation and indications for fault kinematics'.

*We have changed the captions of chapter 4. The former caption "Late Oligocene to recent rifting" was changed into "Structural style of Cenozoic rifting", while the former caption "Late Cretaceous compression" was changed into "Structural style of Late Cretaceous inversion".*

7. Separate the chapter 5. Discuss your interpretation and afterwards precisely write your conclusion. Do not mix!

*We have added a chapter 6. Conclusions*

8. The polyphase evolution of the RVG make some descriptions difficult to follow. Especially in chapter 4 there are plenty descriptions of fault throws and the authors should carefully check their description. E.g. (L359-360) "Due to Cenozoic normal reactivation, only few faults in the study area have net reverse throws as the result of Late Cretaceous compression." If that is the case, how can we ensure that these faults were reactivated? Is there any indication for past fault throws on single faults? How is throw distributed on single faults (e.g. for the pre-Chalk Group strata, syn-inversion strata and rift-strata)? Do the model have the potential to show throw distribution on single faults and for single horizons (e.g. by use of Allan Maps, etc.)? If yes, that would significantly help to illustrate and understand strain distribution across the RVG.

*We use the thickness changes across faults of the Late Cretaceous Chalk Group and Cenozoic syn-rift strata to evaluate their syn-compression and syn-rift throws, respectively.*
*In order to better visualize throw distribution along the FFS, we have added figure 8 that shows the vertical throw of the base of the syn-rift strata along the major faults in the FFS in both northern and southern structural domains.*

9. Furthermore, there are some detailed comments to the text:

    a. L41: What means "both" here? You mention at least three stratigraphic units.

*We have changed this sentence into :"This is indicated by gravimetric maps of the area (Fig. 1) and in more detail in geological maps of the middle Mesozoic (Jurassic), Upper Cretaceous and Cenozoic stratigraphic distributions and thicknesses in the area (c.f. Duin et al., 2006; Deckers et al., 2019)."*

b. L56: Please specify the used data. Is it reflection or refraction seismics? What kind of borehole data was used?

*We have changed this sentence into: "The G3Dv3-model of the area was created by the integration and interpretation of all available 2D seismic reflection and borehole data (borehole descriptions and wireline logs)."*

c. L58: What means "basement" in that context? Crystalline or sedimentary "sub-décollement" strata? I suggest to clarify/define that.

*We have changed the word "basement" into "strata", since the term basement is not relevant for this study.*

d. L70: Here, you mention "Chalk Group". I highly suggest to provide ages and chronostratigrapic names. Please keep in mind that most of your readers are not familar with the local stratigraphic names of the RVG region.

*We have added the chronostratigraphic names Campanian to middle Maastrichtian.*

e. L77-78: Although I understand the intension of this from the modeller's point of view. Nevertheless, it seems a little bit confusing that Mesozoic strata is named 'Cenozoic'. Probably, it would be helpful to modify/enhance the stratigraphic overview figure (Fig. 2) and include some 'real' stratigraphic horizons in relation to your 'model stratigraphy'.

*We have removed the "Cenozoic" from the "Cenozoic pre-rift strata" and now only refer to them as pre-rift strata. This is also mentioned in the text as follows: "For the purpose of this study, the latest Maastrichtian to early Oligocene strata are here referred to as the pre-rift strata".*
*We have now also included real stratigraphic names for the syn-rift strata (Voort, Bolderberg, Diest, etc. formations).*

f. L87: Please use (or define) the names of structures. What is the Roer Valley Rift System? The Roer Valley Graben? Or is the graben a part of that system? You should check the entire manuscript and use unique names.

*These structures are defined in the text of chapter 2.1 as follows:*
*"Major fault activity resumed in the late Oligocene, when the Roer Valley Rift System developed as a northwest-trending branch of the Rhine-Graben-System (Ziegler, 1988), throughout the south-eastern part of the Netherlands, eastern Flanders and adjacent parts of Germany (Fig. 1). This system currently extends over a distance of roughly 200 km and has a width of up to 75 km. Those faults with the strongest displacements divide the central Roer Valley Rift System into the Campine Block in the west, the pre-existing Roer Valley Graben in the center and Peel Block in the east."*

*These different structural features (now also including the Roer Valley Rift System) are also shown in figure 1.*

g. L91: "main faults or those with the largest displacement" - Especially in inverted systems with a high potential to fault reactivation this definition is problematic. Delete that part or provide a definition for 'main faults'.

*We have deleted the word "main".*

h. L92: Please check the consistent use of abbreviations. The use of 'CB vs. Peel Block vs. RVG' in one sentence isn't good style.

*We have changed the abbreviations in this sentence into actual names, so CB to Campine Block and RVG to Roer Valley Graben*

i. L94: What is the Oligocene Voort Formation'? That should be shown in your stratigraphic chart.

*We have added this stratigraphic unit to the stratigraphic chart of figure 2.*

j. L329-330: Contradictory numbers (150m vs. 100m)?

*The vertical throw along the GBF decreases from 250 m towards 150 m, or by 100 m.*

As said above, I strongly suggest the publication of this work. Even if this is still one local piece, such case studies will significantly improve our understanding of inversion tectonics. Furthermore, the study shows how geological modelling can help to understand even complex structures like the RVG and their kinematics. If systematically interpreted and evaluated this third - or, if analysed for various chronostratigraphical horizons as done in that study, fourth - dimension enable various new insights into 'inversion tectonics'. Some modifications and additional work will significantly improve the manuscript. Congratulations for that very interesting contribution. Kind regards, A. Malz

Best regards,
Jef Deckers, Bernd Rombaut, Koen Van Noten and Kris Vanneste

---

## Author Comment (AC3) · 15 Jun 2020

The comment was uploaded in the form of a supplement:
https://se.copernicus.org/preprints/se-2020-23/se-2020-23-AC3-supplement.pdf

---

## Author Response (AR2)

**Dear**

I really enjoyed reading your revised manuscript! Your observation that the fault systems behaved similarly during inversion and later normal reactivation is intriguing. My only suggestions regarding the content of your paper are these:

1. Could you elaborate a bit more on the role of the Carboniferous faults? You cite a literature source for the observation that "large" strike-slip faults are often associated with lithospheric strength contrasts. I guess that is because they juxtapose different types of lithosphere. The Gruitrode Lineament is probably not large enough to do that (or is it?). The fact that it is only present in the footwall of the FFS and abuts the NW-SEstriking faults would suggest to me that its displacement cannot be too large. Are these faults associated with substantial thickness variations of the Carboniferous strata? Or through what other process could they control mechanical strength? 2. I feel that your claim of similar strain but different mechanical strength for the two segments is somewhat contradictory. If strength or rheology somehow relates stress to strain rate, then a weaker region should deform faster and accumulate more strain, provided the stress is similar. Could we argue instead that the southern segment had fewer but weaker faults (and if so, understand why)? If there was a "speed limit" to faults in the northern segment, then more of them were needed to achieve the same strain rate.

I am aware that this amounts to an invitation to speculate, but I think that 's ok as long as it is presented as such.

Two technical things:

In Figs. 3 and 6, the right-hand side must be labeled ENE, not WNW. I. 83 should read "...activity of synsedimentary normal faults with NW-SE to E-W strikes".

Best

regards,

Jonas

Dear Dr. Kley,

Many thanks for your comments on our manuscript! We fully agree with your remarks.

The latest Carboniferous lineaments/faults have fold amplitudes of about 500 m. We have added this in chapter 2.1 on geological background. We agree that this might be too small to consider them as large intra-plate fault systems. It then becomes uncertain if these lineaments coincide with changes in lithospheric strength, which undermines our theory that changes in lithospheric strength are also the cause for the changes in Late Cretaceous and Cenozoic strain distribution in the FFS. Also other mechanisms than changes in lithospheric strengths, such as underlying changes of lithology (for example thickness changes of Early Carboniferous shales that locally represent décollement surfaces) may explain the differences in strain distribution.

Therefore, we have removed the discussion on the lithospheric strength as a possible mechanism. Instead, we limit our discussion to the observation that the change in geometry and strain distribution coincides with underlying pre-existing lineaments. What caused these lineaments is out of the scope of our article but is an intriguing aspect for future research. We have marked the textual changes in yellow shading.

Also thanks for the mentioning of the mistakes on the figures. We have now corrected them.

Best regards,

Jef Deckers, Bernd Rombaut, Koen Van Noten and Kris Vanneste

---

## Author Response (AR3)

Dear Editor,

Many thanks for accepting our manuscript and providing the final remarks! We have modified figure 1 so it now also shows the broader region around the study area (Western Europe).

Best regards,
Jef Deckers, Bernd Rombaut, Koen Van Noten and Kris Vanneste